# Individual Flood Risk Adaptation in Germany: Exploring the Role of Different Types of Flooding

Lisa Dillenardt[1], Annegret H. Thieken[1]

[1]Institute of Environmental Science and Geography, University of Potsdam, Karl-Liebknecht-Strasse 24-25, 14476 Potsdam, Germany

*Correspondence to*: Lisa Dillenardt (dillenardt@uni-potsdam.de)

**Abstract.** Whether and how flood-affected people prepare for flooding is commonly assumed to depend on their perception of the risk, coping options, and responsibilities. Furthermore, the influence of different flood types, i.e., fluvial, flash, and urban pluvial floods, is unclear, but might be relevant for effective risk communication. Up to now, risk communication has mainly addressed fluvial flooding situations. We use survey data from more than 3000 households affected by different types of flooding in Germany to investigate the influence of flood type on adaptive behavior in addition to other influencing

factors. We use descriptive statistics, Kruskal-Wallis tests, and single-factor ANOVA to identify differences and similarities between respondents. We use linear regressions to identify factors that influence households' adaptive behavior in the context of fluvial, pluvial, and flash flooding.

We found that most respondents were motivated to protect themselves, but that there were flood type-specific differences in the factors influencing an adaptive response. For example, those affected by fluvial events had most often implemented

measures before the last flooding and had experienced flooding before, but frequently showed signs of maladaptive thinking and were less likely to implement (more) measures. In contrast, those affected by flash flooding showed less confidence in the effectiveness of measures, but were less likely to rate their costs as too high and were most likely to implement measures after the event. We argue that, inter alia, the severity of the flood processes, the experiences of previous flooding, and the management of flooding, all shape adaptive behavior. Regardless of the type of flooding, the perception of the effectiveness

of adaptive measures and a positive perception of personal responsibility were found to be crucial for motivating those affected to protect themselves. Further analyzes suggest that these two key elements can be strengthened by offering financial support for adaptive measures. We also found that communication on a municipality level enhances residents' sense of personal responsibility. We conclude that communication and management strategies need to involve municipalities and should be tailored to the locally relevant flood type.

**Keywords**: risk communication, protection motivation, flood type, household, survey

## 1 Introduction

Floods were the most damaging climate-related extremes in Europe between 1980 and 2022 (EEA, 2023). To improve flood risk management and reduce flood impacts the European Floods Directive (2007/60/EC) was launched in 2007 in response to several damaging flood events in the European Union (EU) around the year 2000. The directive introduced a structured and integrated flood risk management plan for all EU member states from 2010 onwards, mainly addressing coastal and fluvial floods. In particular, floods that occur due to an overloaded sewage system can be disregarded by member states when adhering to the plan. Germany made use of this option when adapting the Federal Water Act (Wasserhaushaltsgesetz – WHG) in 2009 to the requirements of the Floods Directive (WHG, 2009). Section 72 of the Federal Water Act defines flooding as "[...] a temporary inundation of land not normally covered by water, in particular by surface waters or by seawater entering coastal areas. This does not include flooding from sewage systems." However, in recent years, many German cities have experienced urban pluvial flooding, e.g., the city of Münster in 2014 (Spekkers et al., 2017), Potsdam and Berlin in 2017 and 2019 (Caldas-Alvarez et al., 2022; Dillenardt et al., 2022). Moreover, fast-onset flash floods in the middle hills in May/June 2016 (Laudan et al., 2017; Piper et al., 2016) and July 2021 (Kreienkamp et al., 2021) had huge impacts, i.e., 11 fatalities and €2.6 billion of damage in 2016 and 189 fatalities and €33 billion of damage in 2021 (Thieken et al., 2023). Such impacts from these flood types were unprecedented in the recent past and again called into question current flood risk management approaches.

Integrated flood risk management is built on a variety of risk-reducing measures involving all possible stakeholders, including the general public. Residents in flood-prone areas are obliged to contribute to flood risk reduction as stated in the WHG since 2005. Households can implement Property-level flood risk adaptation measures (PLFRAM) (Attems et al., 2020). These measures cover a wide spectrum of effectiveness and implementation costs and thus range from the creation of emergency plans or the sealing of foundations to the implementation of stationary barriers or relocation to a less at-risk area. PLFRAM can reduce damage caused by floods in-situ in a cost-effective manner (DEFRA, 2008; Hudson et al., 2014; Kreibich et al., 2011; Lamond et al., 2018; Poussin et al., 2015). Using the events of 2013 and 2016 as examples, however, Thieken et al. (2022) illustrated that people have to cope with very different flood pathways in terms of hydraulic characteristics. In addition, different coping and adaptive behaviors were observed (Thieken et al. 2022). Still, explanations and conclusions for risk communication are vague. In view of the devastating event of July 2021 there is an urgent need to better understand peoples' behavior in different (inland) flood settings. To tackle this issue we investigate adaptive behavior of households in the context of three types of flooding: fluvial, flash, and urban pluvial floods, see Figure 1. It should be noted that the distinction between flood types is not always sharply defined and there may be overlaps (Hunt, 2005; Kaiser, 2021; Thieken et al. 2022).

Figure 1: Definitions of the flood types used in this paper based on (Adams et al., 2020; Bruijn et al., 2009; Hunt, 2005; Knocke & Kolivras, 2007; Sweeney, 1992).

All three types of flooding are inland floods, see Figure 1. Inland floods are usually caused by a heavy precipitation or melting event or the sudden release of water due to e.g. dike- or dam breaches (Bruijn et al., 2009; Hunt, 2005). Fluvial floods in particular are caused by overflowing river courses. This can be distinguished from pluvial events, which are more
directly driven by surface runoff and can therefore theoretically occur anywhere (Bruijn et al., 2009). Pluvial floods are triggered by heavy rainfall events or cloudbursts, usually limited in time and space, and which are difficult to predict (DWD, 2016). If pluvial events occur in urban areas with low topography, they are intensified by a high proportion of sealed surfaces and are accompanied by an overload of the sewer and/or drainage system. In this study, we refer to this type of event as urban pluvial flooding. If pluvial events occur in hilly or mountainous terrain, i.e. in steep topography, flash floods
with high flow velocities may occur (Adams et al., 2020; Bruijn et al., 2009). They develop in rather small catchment areas – usually less than six hours after a rain event (Arrow et al., 1995; Knocke & Kolivras, 2007).

To investigate households' adaptive behavior in a structured way, we are using the theoretical frameworks provided by the Protection Motivation Theory (PMT) and the Protection Action Decision Model (PADM). These models identify the appraisals of threat, coping, and responsibility as drivers of adaptive behavior (Lindell & Perry, 2012; Rogers, 1975, 1983).
Various studies have demonstrated the influence of these aspects on the adaptive behavior of households in the context of flooding (Bubeck et al., 2013; Bubeck et al., 2018; Dillenardt et al., 2022; Grothmann & Reusswig, 2006).

The PMT and PADM assume that an individual must first recognize a threat by assessing both its severity (perceived severity) and probability of occurrence (perceived probability). In addition to the threat, the individual will assess the options for coping by estimating the costs and effort required to implement suitable measures (perceived response costs), their
effectiveness in terms of risk reduction (perceived response efficacy), and their own ability to implement these measures

(perceived self-efficacy). The PADM adds to the basic construct of the PMT in that individuals assess the extent to which they themselves (perceived self-responsibility) or public institutions (perceived government responsibility) are responsible for the implementation of measures and widens the understanding of framing/context giving factors (Lindell & Perry, 2012). It is further assumed that if the appraisals of threat, coping, and responsibility are sufficiently high, a motivation to protect oneself (protection motivation) is encouraged, which will then ideally lead to a protective response within the scope of the persons' possibilities. Grothmann and Reusswig (2006) also assume that an assessment of threat that is too low or too high and an assessment of coping strategies that is too low promotes maladaptive thinking or emotional coping mechanisms such as fatalism, denial, procrastination or wishful thinking, of which each is said to have a negative effect on the motivation to protect oneself. Using a hybrid PMT/PADM framework, Dillenardt et al. (2022) found that in the context of urban pluvial flooding, in addition to negative coping mechanisms, negative responsibility appraisal also promotes maladaptive thinking. Another aspect of adaptive behavior is trust in public institutions. Terpstra (2011) found that although trust in public institutions is important for (potentially) affected people to be able to believe the complex hazard assessments of scientists and other stakeholders, trust in public flood protection can also lead to a reduction in their own protection motivation. Currently, this aspect is not well accounted for in the theoretical frameworks. Next to threat and coping appraisals, local flood risk management and previously experienced flooding affect adaptive behaviour also (Kreibich et al., 2005; Poussin et al., 2014; Thieken et al., 2006; Wind et al., 1999).

An examination of the interactions described above between the individual flood types and the factors influencing adaptive behavior leads to a better understanding of flood management strategies and opens up the possibility of tailoring risk communication to the prevailing flood situation in potentially affected areas. In order to close this research gap, this study analyzes survey data from over 3000 households that were affected by fluvial, flash or urban pluvial flooding in Germany and asks: How does the type of flooding influence adaptive behavior? To answer this question, we explore three further research questions:

(1) What adaptive responses were reported by individuals impacted by the three types of flooding?

(2) What factors influenced adaptive behaviour in those affected by the three flood types?

(3) What characteristics of these three groups of respondents explain the differences reported?

## 2. Data & Methods

This study is based on survey data collected via four different survey designs (see Figure 2) between 2014 and 2022 in the course of six surveys among flood-affected households in Germany, see Table 1. While S-1, S-2, S-3, and S-4 were created by a random sampling in affected areas (based on lists of flooded roads; see Thieken et al. (2017)) and considered only

landlines, S-6 was created in Rhineland-Palatinate with the help of the district Ahrweiler, where every third household who had applied for immediate disaster aid was invited to participate. In North Rhine-Westphalia (as well as in S-5) people from the affected areas were invited for a computer-assisted web interview (CAWI) via advertisements on Meta (Facebook and Instagram) and other media. Advertising via Meta to recruit survey participants is a method used in health related research in the last decades (Gilligan et al., 2014; Kapp et al., 2013; Shaver et al., 2019) and have been used by Thieken et al. (2023). A total of 3,670 households were questioned about the impacts of recently experienced flood events along with questions on adaptive behavior based on the PMT and PADM. Data was collected by paper/pencil, computer-assisted web interview (CAWI), and/or computer-assisted telephone interview (CATI), see Figure 2 and Table 1.

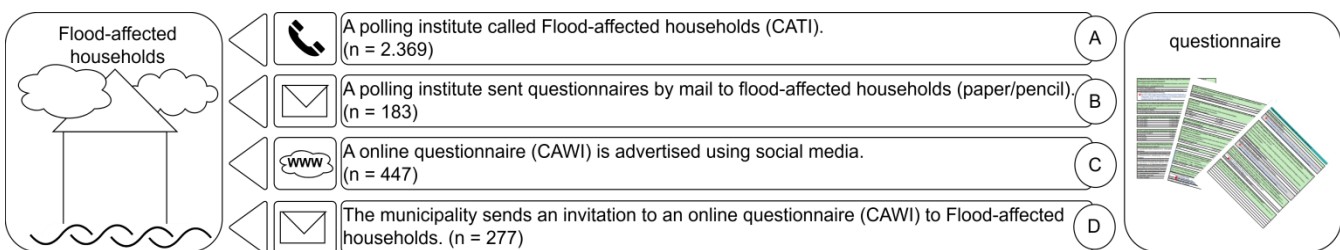

**Figure 2: Simplified illustration of the survey designs A - D used to contact Flood-affected households in Germany**

Data on what PLFRAM were implemented before and after the damaging flood event was collected. However, the questions on PLFRAM were not fully consistent across all surveys due to necessary adaptations to different survey- and event contexts. In order to evaluate the implementation of PLFRAM, all measures were assigned to six main groups based on their principal mode of functioning (Figure 3) as described in current literature (DEFRA, 2008; Hudson et al., 2014; Kreibich et al., 2011; Lamond et al., 2018; Poussin et al., 2015). Table A documents the PLFRAM queried in each survey and their assignment to these six groups. In this Chapter we do not assess the number of PLFRAM implemented, but only whether at least one PLFRAM from a respective group was implemented before or after the flood. It should be noted that this study and the available data cannot clarify the extent to which households adapted appropriately to their local flood situation. This is because the specific PLFRAM or combinations of PLFRAM appropriate to an individual's flood risk depends on many personal and local factors for which no data was collected. On-site visits would be needed for such an evaluation.

For the analysis in this study the respondents of the respective surveys were assigned to the urban pluvial flooding, flash flooding, and fluvial flooding flood types according to the definitions in Figure 1 and based on pathways reported in the survey and further event contexts. Urban pluvial flooding was assigned to respondents affected by pluvial flooding in urban areas with no steep topography and possibly accompanied by overloaded sewer systems as a result of temporally and spatially limited heavy rainfall events. This applies to those affected in the city of Münster and the smaller neighboring city of Greven (Spekkers et al., 2017), as well as to those affected in the cities of Berlin, Potsdam, and Leegebruch (Dillenardt et al., 2022), and 448 surveyed households from S-3 (Thieken et al., 2022). The 53 households affected in the city of

Remscheid are not included in the study, as Remscheids steep topography differs too much from that of the other cities. The respondents to S-1 were assigned to the fluvial flood type, as flooding originated from the rivers Rhine, Weser, Danube, and Elbe (Thieken et al. 2022). In the course of the flooding in June 2013 dike breaches occurred in the federal states of Bavaria

and Saxony-Anhalt (Thieken et al. 2022). Respondents who experienced a dike breach were excluded from the analysis of this paper. Following the classification of (Thieken et al. 2022), we separated from S-3 those who were affected by flash floods and assigned them to the flash flooding flood type, while the remaining cases were considered as urban pluvial flooding. All respondents from S-6 were also assigned to the flash flood type, as this was the primary flood type during the flood of July 2021.

**Table 1: Information on the surveys and demographic information among surveyed households; CAWI: computer-assisted web interview, CATI: computer-aided telephone interviews.**

| No. | Place of flood and survey | Flood type | Responses | Flood event | Survey period | Methods | survey design based on Figure 2 | Publications |
|---|---|---|---|---|---|---|---|---|
| S-1 | More than 160 municipalities across nine federal states | Fluvial flood | 1258 | June 2013 | 18 February – 24 March 2014 | CATI | A | (Thieken et al. 2022) |
|  |  | Levee breach | 394 |  |  |  | A |  |
| S-2 | Münster, Greven | Urban pluvial flooding | 510 | July 2014 | 20 Oct 2015 – 26 Nov 2015 | CATI | A | (Spekkers et al., 2017) |
| S-3 | 67 municipalities in South and West Germany | Urban pluvial flooding | 448 | May – June 2016 | 28. March 2017 – 28. April 2017 | CATI | A | (Laudan et al., 2020; Thieken et al. 2022) |
|  |  | Flash flood | 153 |  |  |  | A |  |
| S-4 | Potsdam, Remscheid, Leegebruch | Urban pluvial flooding | 183 | 2017, 2018, 2019 | 9 July – 9 September 2019 | paper/pencil, CATI | B | (Dillenardt et al., 2022) |
| S-5 | Berlin | Urban pluvial flooding | 115 | 2017, 2018, 2019 | 27 March – 31 May 2020 | CAWI | C | (Berghäuser et al., 2021; Dillenardt et al., 2022) |
| S-6 | North Rhine-Westphalia and Rhineland-Palatinate | Flash flood | 609 | July 2021 | 18 Nov. – 31 Dec. 2022 | CAWI | D | No publication yet |

The demographics of the surveyed households are summarized in Table 2. The reported losses to buildings were corrected for inflation to the year 2022 based on the construction price index (DeStatis, 2023a). The losses to household contents were

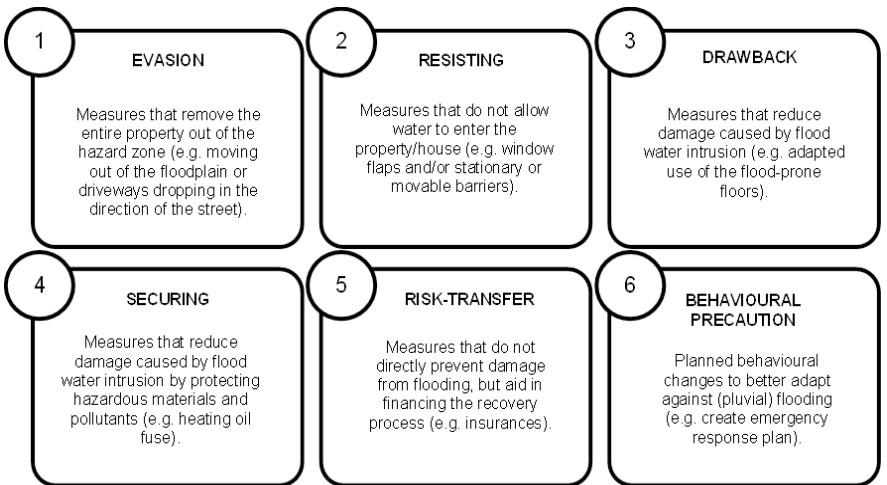

**Figure 2: The six main groups to which the surveyed adaptation measures were assigned; more information about the groups can be found in Table A.**

corrected to the year 2022 based on the consumer price index (DeStatis, 2023b). Regardless of the flood type, more women (57 %) than men (43 %) participated in the surveys. The median age of the respondents was 59, which is approx. eight years above the mean age of the over 18 years old in the German population (DeStatis, 2014). Mainly home or apartment owners participated in the surveys (82 %). On average 2.6 people lived in the households surveyed. More than half of those affected by fluvial flooding reported previous flood experience (62 %), since similar regions had already been affected in August 2002, whereas fewer had such experience among those affected by urban pluvial- (35 %) or flash (21 %) flooding.

We analyzed the data using the statistical software package IBM SPSS 27. To identify significant differences between the three flood types, the Kruskal-Wallis test was performed. For each PMT factor, a Kruskal-Wallis test was first performed for all three flood types. If the Kruskal-Wallis test showed that there was no significant difference between the flood types, this was indicated in Table 4. If the Kruskal-Wallis test showed significant differences, single-factor ANOVAs were performed to better understand identified differences by comparing the flood types in pairs.

Linear regressions were carried out with IBM SPSS 27 to examine in the first step which PMT/PADM factors, i.e., threat, coping and responsibility appraisal, influenced the protection motivation of the respondents. The dependent variable for the regressions presented in Table 6 was protection motivation, which we derived from the items "I will do everything possible to protect myself from flooding" and the item "I would recommend that others take private precautions", see Table A. These two items were combined so that the highest value was always taken for the combined variable. This combined variable enables us to capture protection motivation regardless of whether it relates to the respondent, as in the first item, or to others,

as in the second item. In a second step, the PMT/PADM factors that significantly influenced protection motivation were examined to determine the framing factors that influenced them.

**Table 2: Information on demographic characteristics of surveyed households, classification of gender into people that read their self as female (f), people that read their self as male (m), people that read their self not as female nor as male are counted as "divers" (d).**

| Flood type | Gender m/f/d [%] | Median age [years] | Homeownership [%] | Median monthly net income [€] | Household size (Mean) | Previously experienced floods [%] |
|---|---|---|---|---|---|---|
| Total | 43/57/0.1 | 59 | 79.8 | 2,500 | 2.6 | --- |
| Fluvial N=1258 | 41/59/--- | 62 | 79.7 | 1,750 | 2.4 | 62 |
| Urban Pluvial N=1203 | 43/56/0.3 | 60 | 81.5 | 2,500 | 2.5 | 35 |
| Flash N=762 | 44/56/0.1 | 55 | 76.9 | 3,100 | 3.2 | 21 |


## 3. Results

### 3.1 Comparing the perceived severity of the investigated flood types

In order to characterize the different processes and impacts of the three flood types investigated, key variables are compared in Table 3. Additional data on the perceived flow velocity can be found in Table A. Altogether, the data reveals that flash

floods were particularly severe, since those affected reported the most intense flow velocity, the highest losses to their buildings and building contents, and the highest water depth in their homes, and were most likely to experience floodwater contaminated with fuel oil. In both fluvial flooding and flash flooding about half of those affected had to be evacuated. Flood duration was particularly high in fluvial floods. Inundation indoors, duration, and contamination with fuel oil were lowest for those who had been affected by urban pluvial flooding. The same holds for the financial losses.

**Table 3: Factors used to approximate the severity of the different types of flooding; the reported losses to the building was corrected for inflation to the year 2022 based on the construction price index (DeStatis, 2023a). The losses to the household contents were corrected to the year 2022 based on the consumer price index (DeStatis, 2023b).**

| | total | pluvial | fluvial | flash |
|---|---|---|---|---|
| total number of cases | 3,449 (100 %) | 1,203 (37 %) | 1,258 (39 %) | 762 (24 %) |
| inundation depth indoors [cm] - median | 60 | 20 | 90 | 100 |
| flood duration [h] – median | 60 | 12 | 120 | 24 |
| flow velocity as assessed on a scale from 1 (steadily flowing) to 6 (turbulent flow) - median | --- | 3 | 2 | 5 |
| evacuation [%] | 43 | 6 | 54 | 54 |
| oil contamination [%] | 16 | 2 | 12 | 34 |
| losses to building contents [€] – median | 3,517 | 1,749 | 3,517 | 30,000 |
| losses to building structure [€] - median | 14,627 | 4,343 | 11,251 | 144,780 |

**3.2 Comparison of the measures taken by those affected before and after a perceived flood**

Figure 4 shows the share of surveyed households that implemented at least one PLFRAM from a given category (see Fig. 3 and A) of PLFRAM before (Figure 4, left) and/or after a flood, see Figure 4, middle. The results of before and after are summed up in Figure 4, right.

Those affected by fluvial flooding in 2013 had implemented PLFRAM most frequently before the event and very few measures after the event, while those affected by flash floods (in 2016 and 2021) had rarely implemented PLFRAM before 210 the event, but frequently after the event. Those who were affected by fluvial or flash floods had taken out insurance before the last flood event in roughly equal numbers and more often than those affected by urban pluvial flooding. After the event, those affected by flash flooding were particularly likely to take out insurance, making them the most likely group for this kind of PLFRAM. After the event, roughly the same number of those affected by urban pluvial flooding and flash floods had implemented measures in the categories "Resistance" and "Drawback". Those affected by flash floods implemented 215 measures in the category "Securing" more frequently after they had been flooded.

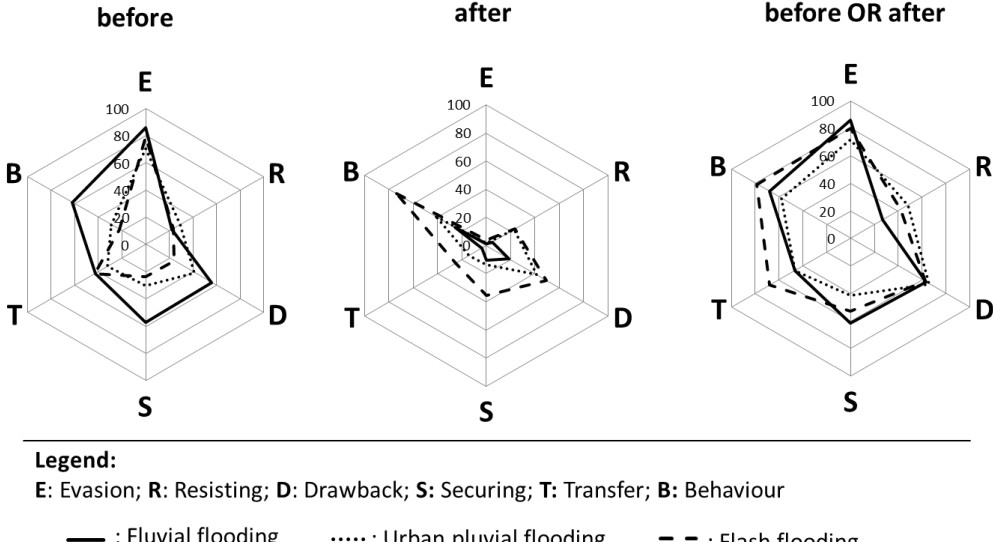

**Figure 3: Proportion of respondents per flood type who implemented at least one measure from a PLFRAM group before and/or after a flood; further information on the PLFRAM groups can be found in Fig. 2 and Table A.**

Considering together the PLFRAM implemented before and after the event, a pattern can be seen across the flood types. Preparedness measures were implemented quite frequently. Evasion measures were predominantly implemented before the most recent flood event. Drawback measures were implemented before and after with somewhat equal frequency by 60 % of

respondents. In addition to the above-mentioned similarities it is striking that those affected by fluvial flooding less frequently implemented resistance measures.


### 3.3 Comparing potential drivers of adaptive behaviour

Table 4 compares the flood types in terms of respondents' attitudes towards adaptation to flood risk on the theoretical basis of the PMT and PADM, using Kruskal-Wallis tests and, if Kruskal-Wallis tests indicates differences, single-factor ANOVA.

Table 5 shows the median and mean values of each item analyzed. More detailed information on the answers to the items can be found in Table B. Percentages show the proportion of respondents who selected either one and two or five and six on a scale from one to six and is derived from the data presented in Table B.

With regard to threat appraisal, respondents rate the severity of a future flood as high (median values of 5 for fluvial, 4 for pluvial, and 5 for flash floods on a scale from 1 – not bad, to 6 – very bad), but often do not believe that such a future event

will affect them (median values of 5 for fluvial, 3 for pluvial, and 4 for flash floods on a scale from 1 – unlikely, to 6 – likely). The group of those reporting a high perceived severity is comparable and larger among those affected by fluvial or flash flooding than among those affected by urban pluvial floods. As for the rating of the probability of a future event, Table 5 shows a gradient from those affected by urban pluvial flooding, who rate the probability of a future event the lowest (median: 3), to those affected by flash flooding (median: 4) and those affected by fluvial flooding, who rate the probability of being affected again the highest (median: 5).

being affected again the highest (median: 5).

Coping appraisal is investigated by looking at perceived self-efficacy, perceived response efficacy, and the perceived response cost. Self-efficacy is rather high for around 60 % of respondents, and comparable across all samples and flood types, indicating that self-efficacy is person-related rather than event- or flood type related. Most of those affected by urban pluvial and fluvial flooding tend to have a high and comparable response efficacy (median: 5), while this proportion is lower

for those affected by flash floods (median: 4). About 60% of those affected by urban pluvial floods and 56 % of those affected by flash floods perceive the response costs as (too) high and are comparable in this respect, while this proportion is higher for those affected by fluvial floods (69 %).

Self-responsibility is perceived as high by all respondents. However, the level of self-responsibility is higher among those affected by fluvial (median: 6) than among those affected by urban pluvial or flash flooding (median: 5). At the same time,

those affected by fluvial, urban pluvial or flash floods believe that public institutions have a responsibility to implement flood protection measures (median: 4). However, only flash and pluvial flooding are comparable here, see Table 4, and the mean values in Table 5 reveal that those affected by fluvial flooding stand out in seeing public institutions as slightly more responsible. Yet, most of those affected by flooding (median: 1-2) have little confidence in public flood protection measures. Moreover, most people affected by flooding have little confidence in state financial aid (median: 2-3).

In general, most respondents believe that there is enough information available about flooding and flood adaptation (median: 3). However, fewer respondents affected by urban pluvial (median: 3) and flash flooding (median: 2) believe that

**Table 4: Results of Kruskal-Wallis and ANOVA post-hoc tests[a]: significance values are adjusted by Bonferroni correction for multiple tests; count = count of cases used for this analysis; more details about items can be found in Table A; \*\*\* p<0.01, \*\* p<0.05, \* p<0.1, "STS" = standardized test statistics .**

| Item | H-Test | ANOVA (pair-wise) | | |
| --- | --- | --- | --- | --- |
| | | Fluvial versus pluvial | Fluvial versus flash | Flash versus pluvial |
| Perceived probability | Count: 2856 | STS: 16.363 | STS: 11.204 | STS: -3.621 |
| | Test statistic: 279.741*** | Ad. Sig[a].: 0.000 | Ad. Sig[a].: 0.000 | Ad. Sig[a].: 0.001 |
| Perceived severity | Count: 2641 | STS: -13.400 | STS: -0.589 | STS: 12.801 |
| | Test statistic: 248.531*** | Ad. Sig[a].: 0.000 | Ad. Sig[a].: 1.000 | Ad. Sig[a].: 0.000 |
| Perceived self-efficacy | Count: 2634 | Retain null hypothesis | | |
| | Test statistics: 1.686 | | | |
| Response efficacy | Count: 2829 | STS: 0.460 | STS: -6.878 | STS: -7.610 |
| | Test statistics: 66.584*** | Ad. Sig.[a]: 1.000 | Ad. Sig.[a]: 0.000 | Ad. Sig.[a]: 0.000 |
| Response costs | Count: 2620 | STS: -5.916 | STS: -5.128 | STS: 0.129 |
| | Test statistics: 41.416*** | Ad. Sig.[a]: 0.000 | Ad. Sig.[a]: 0.000 | Ad. Sig.[a]: 1.000 |
| Responsibility government | Count: 2782 | STS: -3.820 | STS: -2.464 | STS: 0.957 |
| | Test statistics: 15.058*** | Ad. Sig.[a]: 0.000 | Ad. Sig.[a]: 0.041 | Ad. Sig.[a]: 1.000 |
| Responsibility self | Count: 2804 | STS: -10.447 | STS: -14.891 | STS: -5.917 |
| | Teststatist.: 235.118*** | Ad. Sig[a].: 0.000 | Ad. Sig.[a]: 0.000 | Ad. Sig.[a]: 0.000 |
| Trust – public flood protection | Count: 2804 | STS: 10.780 | STS> -5.088 | STS: -15.203 |
| | Test statistics: 256.027*** | Ad. Sig[a].:0.000 | Ad. Sig.[a]: 0.000 | Ad. Sig.[a]: 0.000 |
| Trust – financial aid | Count: 2008 | STS: 0.432 | STS: -5.643 | STS: -6.460 |
| | Test statistics: 47.959*** | Ad. Sig.[a]:1.000 | Ad. Sig.[a]:0.000 | Ad. Sig.[a]:0.000 |
| Financial support | Count: 2008 | STS: 0.432 | STS: -5.643 | STS: -6.460 |
| | Test statistics: 47.959*** | Ad. Sig.[a]: 1.000 | Ad. Sig.[a]: 0.000 | Ad. Sig.[a]: 0.000 |
| Protection motivation self | Count: 2779 Test statistics:319.338*** | STS: -10.931 | STS: -17.674 | STS: -8.623 |
| | | Ad. Sig.[a]:0.000 | Ad. Sig.[a]: 0.000 | Ad. Sig.[a]:0.000 |
| Protection motivation others | Count: 2796 | STS: -7.888 | STS: -5.246 | STS: 1.768 |
| | Test statistics: 66.818*** | Ad. Sig.[a]: 0.000 | Ad. Sig.[a]: 0.000 | Ad. Sig.[a]: 0.231 |
| Emotional coping –fatalism | Count: 2862 Test statistics:102.101*** | STS: -8.171 | STS: -9.187 | STS: -2.086 |
| | | Ad. Sig.[a]:0.000 | Ad. Sig.[a]:0.000 | Ad. Sig.[a]:0.111 |
| Emotional coping – denial | Count: 2863 Test statistics:378.274*** | STS: -15.409 | STS: -17.880 | STS: -4.565 |
| | | Ad. Sig[a].: 0.000 | Ad. Sig.[a]:0.000 | Ad. Sig.[a]:0.000 |
| Information – general | Count: 2524 | Retain null hypothesis | | |
| | Test statistics: 4.637* | | | |
| Information - municipalities | Count: 2636 | STS: -11.489 | STS: -14.061 | STS: -4.011 |
| | Test statistics: 225.746*** | Ad. Sig[a].: 0.000 | Ad. Sig.[a]: 0.000 | Ad. Sig.[a].: 0.000 |

**Table 5: Items asked in the surveys with the scales used and the answers per flood type (median and mean).**

| No. | topic | item asked in survey | scale | fluvial | pluvial | flash |
|---|---|---|---|---|---|---|
| | | | | data presented as median (above) and mean (below). | | |
| 1 | perceived probability | "How likely do you think it is that your apartment or house will be hit by flooding again?" | 1 - very unlikely | 5 | 3 | 4 |
| | | | 6 - very likely | 4.6 | 3.4 | 3.6 |
| 2 | perceived severity coding reversed | "How bad do you expect the consequences of a future event will be?"[1] | 1 –not bad | 5 | 4 | 5 |
| | | | 6 – very bad | 4.5 | 3.8 | 4.5 |
| 3 | pesponse efficacy coding reversed | "Adaptive measures can significantly reduce flood damage." | 1 - I do not agree | 5 | 5 | 4 |
| | | | 6 - I fully agree | 4.4 | 4.5 | 3.9 |
| 4 | pesponse cost | "Adaptive measures are far too expensive." | 1 - I fully agree | 3 | 3 | 3 |
| | | | 6 - I do not agree | 2.9 | 3.4 | 3.3 |
| 5 | self-efficacy | "Personally, I do not feel able to implement any of the measures mentioned above." | 1 - I fully agree | 5 | 5 | 5 |
| | | | 6 - I do not agree | 4.3 | 4.3 | 4.4 |
| 6 | responsibility public coding reversed | "Flood prevention is the responsibility of public institutions and not of private individuals." | 1 - I do not agree | 4 | 4 | 4 |
| | | | 6 - I fully agree | 4.0 | 3.8 | 3.9 |
| 7 | responsibility self coding reversed | "Every individual has a responsibility to reduce flood damage as much as possible." | 1 - I do not agree | 6 | 5 | 5 |
| | | | 6 - I fully agree | 5.3 | 4.7 | 4.2 |
| 8 | fatalism coding reversed | "There is generally nothing that can be done about flooding and flood damage." | 1 - I do not agree | 4 | 4 | 3 |
| | | | 6 - I fully agree | 4.1 | 3.5 | 3.3 |
| 9 | denial coding reversed | "I don't even want to think about future flood damage!" | 1 - I do not agree | 6 | 5 | 4 |
| | | | 6 - I fully agree | 5.2 | 4.2 | 3.8 |
| 10 | trust coding reversed | "The flood protection in our region is so good, I don't need to take private adaptation measures." | 1 - I do not agree | 1 | 2 | 1 |
| | | | 6 - I fully agree | 2.0 | 2.7 | 1.6 |
| 11 | public support coding reversed | "There are enough tax concessions and subsidy programs for financing adaptive measures." | 1 – I do not agree | 2 | 3 | 2 |
| | | | 6 - I fully agree | 2.7 | 2.8 | 2.2 |
| 12 | information available | "There is far too little information and advice on private flood prevention." | 1 - I fully agree | 3 | 3 | 3 |
| | | | 6 - I do not agree | 3.4 | 3.3 | 3.2 |
| 13 | local information coding reversed | "Our municipality provides very good information about flood risks and possible precautionary measures." | 1 - I do not agree | 4 | 3 | 2 |
| | | | 6 - I fully agree | 3.8 | 2.9 | 2.5 |
| 14 | protection motivation | "Personally, I will do everything I can to protect the house I live in from flooding." | 1 – I do not agree | 6 | 5 | 5 |
| | | | 6 - I fully agree | 5.4 | 4.9 | 4.3 |
| 15 | coding reversed | "I would recommend that others implement adaptive measures" | 1 - I do not agree | 6 | 5 | 6 |
| | | | 6 - I fully agree | 5.2 | 4.8 | 4.8 |

---

[1] As S-1 was designed as a panel survey and this item was not asked in the first wave of the survey, the results for this item are based on the results of the 2nd wave of the panel survey, in which n = 710 households from the 1st wave took part.

there is enough local information available from the municipalities. Those affected by fluvial floods stand out here, as they tend to feel better informed by their municipalities (median: 4).

Regardless of the type of flooding, over 70 % of respondents have a rather high motivation to protect themselves and/or would recommend others to do the same. A gradient can be seen in the motivation to protect oneself (fluvial - median: 5.5, pluvial - median: 4.9 and flash - median: 4.3, see Table 5). The proportion of respondents showing signs of fatalism is higher
among those affected by fluvial and urban pluvial (median: 4) than by flash (median: 3) flooding. The proportion of respondents showing signs of denial is high among those affected by fluvial flooding (median: 6) and less high among those affected by urban pluvial and flash flooding (median: 4-3). Hence, the group of those affected by fluvial flooding demonstrates that high protection motivation and emotional coping are not mutually exclusive.

3.4**. Results of regression analyses**

The PMT/PADM aspects analyzed in this study and how they affect protection motivation were tested in regressions 1 – 3, see Table 6. After identifying the PMT/PADM aspects that showed significant influences in the respective flood type contexts, we investigated which framing factors influence them, see Table 7. This table shows the dependent variables of all
linear regressions in the second row, "dependent variables". "Perceived flood inundation/velocity" corresponds to item No. 4, Table 1, "An average person could have stood [...]". In all linear regressions presented here, the flood types were considered, meaning that the datasets presented in Table 2 were used.  Each column in Tables 6 and 7 represents a linear regression. All PMT/PADM aspects examined show significant influences for at least one type of flooding, which hints on the suitability of the PMT/PADM based hybrid framework to start a discussion on the factors influencing protection motivation in the context
of flooding. Only significant correlations are discussed hereafter. Correlation coefficients are shown in brackets.

The influence of threat appraisal on protection motivation is examined through the perception of the flood inundation/velocity of the last flooding and the perceived probability of future flooding. Protection motivation is negatively linked to perceived flood inundation/velocity for those affected by fluvial flooding (-0.131) and flash flooding (-0.128). Hence, a higher perceived severity of the last event may inhibit protection motivation. However, only for those who have
been affected by urban pluvial flooding, financial losses are positively linked to protection motivation (3.386E-6). This link appears to be minimal because this item is not on a scale from 1 to 6. Instead this item captures the overall loss in Euro. Therefore, the financial loss experienced seems to trigger protection motivation only if high losses have been experienced. The link between the perceived probability of a future event and protection motivation in the context of pluvial flooding is positive but minimal (0.096).
The influence of coping appraisal on protection motivation is analyzed through perceived response efficacy, perceived self-efficacy, and perceived response cost. Perceived response efficacy is highly significant across all types of flooding, see Table 6, and thus influences the protection motivation regardless of flood type. Perceived self-efficacy positively influences the protection motivation of those affected by fluvial (0.083) or flash (0.196) flooding, which is in line with PMT. Response

costs are positively linked to protection motivation in fluvial flooding (0.066) and in the context of pluvial flooding (0.097); however, those linkages are minimal.

**Table 6: Results of regression analysis; dependent variable for all four regressions is the protection motivation of households; standard errors in parentheses; significance indicated as follows: \*\*\* p<0.01, \*\* p<0.05, \* p<0.1**

| | regression 0 general (n=3,449) | regression 1 fluvial (n=1,258) | regression 2 pluvial (n=1,203) | regression 3 flash (n=762) |
|---|---|---|---|---|
| | dependent variable: protection motivation - coding reversed | | | |
| constant | 1.653*** | 2.625*** | 1.100** | 0.744 |
| | (0.282) | (0.456) | (0.379) | (0.048) |
| financial loss | -3.094E-7* | -1.153-6 | 3.386E-6** | -6.727E-8 |
| | (0.000) | (0.000) | (0.000) | (0.000) |
| perceived flood inundation/velocity | -0.144*** | -0.131* | 0.068 | -0.128* |
| | (0.044) | (0.073) | (0.123) | (0.076) |
| Perceived probability of future floods | 0.019 | -0.052 | 0.096* | -0.009 |
| | (0.026) | (0.042) | (0.051) | (0.048) |
| perceived response efficacy coding reversed | 0.252*** | 0.240*** | 0.255*** | 0.228*** |
| | (0.026) | (0.038) | (0.047) | (0.051) |
| perceived response cost | 0.064** | 0.066* | 0.097** | 0.064 |
| | (0.025) | (0.039) | (0.048) | (0.047) |
| perceived self-efficacy | 0.089*** | 0.083** | 0.013 | 0.196*** |
| | (0.026) | (0.037) | (0.048) | (0.052) |
| perceived government responsibility coding reversed | 0.041 | -0.026 | 0.109** | 0.128** |
| | (0.027) | (0.0039) | (0.053) | (0.051) |
| perceived self-responsibility coding reversed | 0.273*** | 0.246*** | 0.238*** | 0.269*** |
| | (0.030) | (0.054) | (0.058) | (0.048) |
| R-squared | 0.243 | 0.177 | 0.234 | 0.262 |

The influence of responsibility appraisal on protection motivation is analyzed through perceived self-responsibility and perceived government responsibility. A positive linkage between perceived self-responsibility and protection motivation is found across all types of flooding, see Table 6. Thus, the assessment of self-responsibility influences the protection motivation, regardless of the type of flooding. Protection motivation is positively linked to government responsibility for

those affected by pluvial (0.109) and flash (0.128) flooding. In conjunction with the positive influence of a sense of personal responsibility, communicating responsibilities in general may positively affect the motivation to adapt.

The PMT factors identified as significant in Table 6 were then analyzed to determine the extent to which they were influenced by framing factors. However, the framing factors analyzed can only be a starting point for investigating the influences of framing factors and are limited to those included in the surveys. In order to see to what extent event-specific and thus survey-specific factors could influence the PMT factors, we worked with dummy variables in these analyzes. For this purpose we created a dummy variable for each survey from Table 2 and implemented these variables in the linear regressions of the urban pluvial and flash flooding types. As the data on fluvial flooding originates from one survey, no dummy variables were created for this type of flooding. Only significant results are presented below. All results can be found in Table 7. The financial loss incurred and the flood inundation/velocity were not investigated. R-squared is generally lower than in the regression analyzes of the PMT factors. This indicates that the independent variables do not yet include all framing factors that would reveal influences in these contexts. Nevertheless, the incomplete list of framing factors is used to identify meaningful relationships between PMT/PADM aspects and framing factors.

The event-specific dummy variables improve the R-squares of the regression models and capture influences that distinguish those affected by a particular event from others affected by the same type of flood. These event-specific effects can be time-, survey- or location-specific, although it is impossible to break this down precisely based on our data. The perceived probability of a future event is lower among those who experienced urban pluvial flooding events in Berlin, Potsdam and Leegebruch. The response efficacy was lower among those affected by urban pluvial flooding in 2016. The residents of Berlin stand out as perceived government responsibility as lower. Regarding flash flooding, those affected by the event in 2021 stand out with a high perceived self-efficacy, while those affected in 2016 perceive  perceived government responsibility as high. However, it must be mentioned at this point that we cannot separate what causes these differences on the basis of our data. It therefore remains open to interpretation and discussion as to whether these are local aspects or aspects specific to the survey. Further research is needed in this respect.

Increasing age is negatively linked to respondents' self-efficacy in the context of fluvial (-0.030) and flash (-0.019) flooding. Older people, therefore, tend to feel unable to implement PLFRAM. Increasing age is negatively linked to perceived self-responsibility in the context of flash flooding (-0.009), which indicates that self-responsibility assessment decreases with increasing age. Confidence in public flood defences is negatively related to the perceived response costs for pluvial flooding (-0.137), which indicates that people with a high level of confidence in public flood defences tend to rate the costs of PLFRAM as (too) high. When respondents have high trust in public flood defences, they show a higher perceived self-efficacy in the context of fluvial floods (0.099), but a lower perceived self-responsibility in the context of fluvial (-0.102) and pluvial (-0.100) floods. The overall picture suggests that trust in public flood protection can be a rather hindering factor in promoting adaptive behavior.

A positive perception of the availability of financial support increases perceived response efficacy in the context of fluvial (0.173) and flash (0.104) flooding. In addition, a positive link between the perception of financial aid and perceived self-responsibility in the context of pluvial (0.166) and flash (0.126) flooding. Both perceived response efficacy and perceived self-responsibility were identified as the clearest triggers of protection motivation in the analysis, see Table 6. Since the perceived availability of financial aid enhances them, communicating financial aid may be crucial to support the implementation of PLFRAM. We further found a negative link between the perceived availability of financial aid and the perceived government responsibility in the context of flash flooding.

Availability of general information has been shown to positively influence perceived response efficacy in the context of flash floods (0.116), perceived response costs positively in the context of fluvial (0.222) and pluvial (0.380) floods, and self-efficacy in the context of fluvial floods (0.135) and flash floods (0.249). Availability of general information has been shown to negatively influence self-responsibility (-0.089) in flash floods and government responsibility in relation to pluvial floods (-0.203) and flash floods (-0.273). The overall picture thus shows that a positively perceived availability of general information can promote adaptive behavior in which those affected see the government as less responsible and, at the same time, assess the costs and feasibility of measures more positively. While the availability of general information impacts the perception of the government's responsibility, it is information from the municipalities that might promote the perception of personal responsibility among the respondents, at least in the context of fluvial (0.092) and urban pluvial (0.088) flooding. However, in the context of pluvial flooding, the availability of local information links negatively with one perceived probability of a future event (-0.075), which might be a hint that it is challenging to communicate occurrence probabilities as suggested in literature (Grounds et al., 2017).

There is a positive connection between ownership and perceived response efficacy (0.179) and perceived response cost (0.234) in the context of pluvial flooding. In the context of flash flooding ownership links positively with perceived self-efficacy (0.295) and negatively with one's perceived government responsibility (-0.175). Ownership further links positively with perceived self-responsibility in the context of pluvial (0.167) and flash (0.236), but negatively in the context of fluvial (-0.152) flooding. Hence, homeowners affected by flash flooding tend to rather not see the government as responsible but themselves. Previously experienced floods positively affect the perceived probability of a future event occurring in the context of pluvial flooding (0.254). The flood experience positively affects the perception of response efficacy in the context of flash (0.153) flooding.

**Table 7: Results of regression analysis for those affected by fluvial flooding; dependent variables (first line) are those variables of TABLE 6-fluvial (column 3) that are significant; standard errors in parentheses, significance is indicated as follows: \*\*\* p<0.01, \*\* p<0.05, \* p<0.1**

| | | threat appraisal | coping appraisal | | | | | | | responsibility appraisal | | | | |
|---|---|---|---|---|---|---|---|---|---|---|---|---|---|---|
| Dependent variables | | perceived probability | perceived response efficacy | | | perceived response cost | | perceived self-efficacy (reverse scale) | | perceived self-responsibility | | | perceived government responsibility | |
| data set, compare Table 2 | | pluvial | fluvial | pluvial | flash | fluvial | pluvial | fluvial | flash | fluvial | pluvial | flash | pluvial | flash |
| | constant | 4.104*** (0.400) | 0.968* (0.553) | 3.628*** (0.418) | 3.077*** (0.500) | 1.515** (0.533) | 1.395*** (0.451) | 4.963*** (0.554) | 3.636*** (0.531) | 2.456*** (0.379) | 3.942*** (0.412) | 2.251*** (0.495) | 4.743*** (0.414) | 5.316** (0.381) |
| | dummy_S-2 | excluded | --- | excluded | --- | --- | excluded | --- | --- | --- | excluded | --- | excluded | --- |
| | dummy_S-3 | 0.101 (0.144) | --- | -0.316* (0.161) | excluded | --- | -0.020 (0.155) | --- | excluded | --- | -0.163 (0.143) | excluded | -0.016 (0.153) | 0.550** (0.177) |
| | dummy_S-4 | -2.092*** (-0.032) | --- | 0.460 (0.468) | --- | --- | -0.393 (0.017) | --- | --- | --- | -0.206 (0.412) | --- | -0.612 (0.428) | --- |
| | dummy_S-5 | -1.563*** (0.276) | --- | 0.418 (0.279) | --- | --- | 0.209 (0.276) | --- | --- | --- | -0.242 (0.271) | --- | -0.586** (0.290) | --- |
| | dummy_S-6 | --- | --- | --- | 0.288 (0.222) | --- | --- | --- | 0.720** (0.229) | --- | --- | 0.070 (0.209) | --- | excluded |
| | age | -0.003 (0.004) | 0.006 (0.006) | 0.001 (0.004) | -0.002 (0.005) | -0.003 (0.005) | 0.004 (0.004) | -0.030*** (0.006) | -0.019** (0.074) | -0.001 (0.004) | 0.001 (0.004) | -0.009* (0.005) | -0.001 (0.004) | -0.001 (0.005) |
| independent variabales | trust in public flood protection **coding reversed** | 0.008 (0.038) | 0.024 (0.058) | 0.040 (0.043) | -0.013 (0.075) | 0.088 (0.056) | -0.137** (0.047) | 0.099* (0.058) | -0.058 (0.074) | -0.102** (0.040) | -0.100*** (0.036) | 0.022 (0.073) | 0.037 (0.040) | 0.082 (0.060) |
| | availability of financial aid **coding reversed** | -0.042 (0.041) | 0.173*** (0.050) | 0.066 (0.046) | 0.104* (0.059) | 0.069 (0.048) | 0.042 (0.047) | 0.034 (0.050) | -0.057 (0.057) | -0.001 (0.034) | 0.116** (0.041) | 0.126** (0.056) | -0.008 (0.045) | -0.106* (0.056) |
| | availability of general information | -0.014 (0.039) | 0.023 (0.044) | -0.025 (0.044) | 0.116** (0.047) | 0.222*** (0.043) | 0.380*** (0.041) | 0.135** (0.044) | 0.249*** (0.044) | -0.009 (0.030) | -0.010 (0.037) | 0.089* (0.049) | -0.203*** (0.040) | -0.273** (0.051) |
| | availability of local information **coding reversed** | -0.075* (0.039) | 0.043 (0.045) | 0.049 (0.047) | 0.065 (0.053) | 0.024 (0.043) | 0.007 (0.042) | -0.063 (0.045) | -0.052 (0.052) | 0.092** (0.031) | 0.088** (0.036) | 0.076 (0.055) | -0.020 (0.042) | -0.044 (0.051) |
| | ownership | -0.062 (0.092) | 0.000 (0.099) | 0.179* (0.094) | -0.005 (0.088) | -0.036 (0.095) | 0.234** (0.098) | 0.075 (0.098) | 0.295** (0.093) | -0.152** (0.067) | 0.167* (0.090) | 0.236** (0.092) | -0.018 (0.042) | -0.175* (0.098) |
| | flood experience | 0.254*** (0.065) | 0.045 (0.055) | -0.019 (0.073) | 0.153* (0.086) | 0.000 (0.054) | 0.061 (0.070) | 0.061 (0.056) | 0.133 (0.088) | -0.028 (0.038) | 0.050 (0.066) | 0.100 (0.006) | 0.010 (0.067) | -0.006 (0.087) |
| | R-squared | **0.098** | 0.039 | **0.033** | **0.044** | 0.074 | **0.167** | 0.095 | **0.129** | 0.044 | **0.057** | 0.068 | **0.068** | **0.127** |

**4 Discussion**

**4.1. Adaptive responses to the different flood types**

At 95.6 %, most respondents (98.6 % of those affected by fluvial, 97.5 % of those affected by flash, and 91.5 % of those affected by urban pluvial flooding) had implemented at least one PLFRAM before or after the damaging event regardless of flood type. This reflects the generally progressed adaptation of those affected by flooding and the boost in adaptation after

380 damaging events. Respondents were particularly likely to adapt their behavior by e.g. seeking information, attending seminars and neighbourhood assistance meetings, creating emergency plans, or other preparatory measures (e.g. procuring pumps). This is consistent with the findings of Grothmann and Reusswig (2006), who found that searching for flood-related information is the most frequently performed adaptation. These positive attitudes towards preparedness measures do not directly reduce future damage, but demonstrate the need for information after a flood.

Evasion measures were very rarely implemented after an event, see Figure 4.4. Since measures in this group are difficult to implement subsequently, such as making driveways drop towards the road, and also require a great effort, such as moving to a less flood-prone area, it is likely that these measures undergo individual cost-benefit assessments. They are much easier to be implemented when planning or constructing a home and should thus be communicated to people involved in construction projects. In contrast, the possibility of taking out insurance could be communicated before and after events. Communicating

on this topic is likely to have an impact. In Germany, mandatory flood insurance has been discussed since the devastating floods of 2002 (Thieken et al., 2006). Market penetration has increased from 19 % in 2002 to 49 % in 2021 (GDV, 2022) and to 52 % in 2022 (GDV, 2023). Furthermore, our data show that the uptake of insurance policies covering flood losses before the last event was around 40 % among all households surveyed. Insurance was purchased after flooding especially by those who were affected by flash flooding. This does make sense, since the amount of flood losses by flash floods is very

high (see Table 3) and people with insurance can in general rely on loss compensation based on the insurance contract.

**4.2 Appraisal of threat, coping, and responsibility in the context of different flood types**

The appraisal of threat is assumed to be a crucial driver in the PMT and PADM. It is formed by the perceived severity and

400 perceived probability (of a future event) and is expected to influence protection motivation positively, if it is not too high (Grothmann & Reusswig, 2006; Lindell & Perry, 2012; Prentice-Dunn & Rogers, 1986). Fluvial floods were perceived as more devastating than urban pluvial floods but less devastating than flash floods. Hence, our analyzes illustrate that the flood types examined were perceived very differently by the respondents. These perceptions are confirmed by research on events that were not analyzed in this study. Poussin et al. (2014) found for flooding in France that fluvial floods caused less damage

and fewer fatalities than flash floods, and Spekkers et al. (2017) observed rather minor water depth during an urban pluvial flood in Amsterdam and did not report any fatalities. In contrast to flash flooding, the urban pluvial events were perceived as the least severe in our data.

Those who were affected by fluvial floods report both a higher perceived severity and a higher perceived probability of future flooding, which might also be due to repeatedly experienced flooding of this type. Since this group had also implemented the most PLFRAM before the event, our data do not allow us to observe the negative feedback loop between the implementation of PLFRAM and the appraisal of threat that was described by Bubeck et al. (2012) and confirmed by Poussin et al. (2014) for the context of fluvial events. Our data suggest that the implementation of PLFRAM in the past did not lower the respondents' assessment of the threat, or that the assessment of the threat, which may have decreased after the implementation of PLFRAM, increased again after experiencing another flood. However, in this context, the fact that those who were affected by fluvial flooding retrofitted less PLFRAM after the last flood event may indicate that if those affected by floods implement PLFRAM and then experience flooding and losses again, their higher risk assessment may lead not to the implementation of more PLFRAM, but rather to higher maladaptive thinking, as maladaptive thinking was particularly pronounced in those affected by fluvial flooding.

The regression analysis of PMT/PADM aspects, see Table 6, reveals no significant link between perceived probability of a future event and protection motivation for fluvial and flash flooding, what is in line with findings in Australia (Bird et al., 2013). For all respondents, the mean of the perception of a floods' severity is higher than its perceived probability, indicating that many of those affected are aware that flooding can cause high levels of losses, but that they themselves might not be affected by it (again), which is in line with findings of Netzel et al. (2021) in the context of urban pluvial flooding. Communicating the probability of future events occurring in a particular locality may therefore be a possibility to enhance one's local risk awareness. Return periods may not be the most suitable tool here (Grounds et al., 2018), since they suggest long time periods between flood events if not well explained. Perceived inundation/velocity showed an effect that decreased the motivation to protect oneself in the context of fluvial and flash floods, which were perceived as more severe. In the context of urban pluvial flooding the financial loss triggers protection motivation if those losses have been high. Future information campaigns for urban pluvial flooding could therefore communicating high losses, if those are to be expected in a specific area.

In addition to the assessment of the threat, it is the assessment of coping options that shapes adaptive behavior and is perhaps even the stronger driving force here (Poussin et al., 2014). Therefore, it is a positive aspect that most respondents after an event tend to believe that PLFRAM reduce flood damage and that they can implement these measures, and thus generally tend to perceive both a high sense of self-efficacy and a high perceived response efficacy. Perceived response efficacy has been found to positively influence protection motivation regardless of flood type. The fact that those affected by flash flooding have a lower perceived response efficacy but also less often perceive the costs of measures as too high may hint that this group of respondents experienced particularly severe flooding, which undermined the effectiveness of many PLFRAM and put their costs into perspective. Therefore cost-benefit analyzes of PLFRAM could be carried out on a flood type-

specific basis and communicated to those who may be potentially affected. For urban pluvial flood events in particular, it should be investigated which PLFRAM can reduce the expected damage in a cost-effective manner, since floods of this type are characterized as less severe (see Table 3) and response costs are perceived as rather high by those affected, see Table 5. Often, only small changes to the buildings, e.g., the implementation of ground sills, might already help prevent water from entering the building.

Responsibility appraisal is expected to positively influence protection motivation. This study divides responsibility appraisal into one's own perceived responsibility and the perception of the government's responsibility. From the regression analyzes, we know that self-responsibility has a positive effect on protection motivation, regardless of the type of flooding, and that perceived government responsibility in the context of pluvial and flash flooding also has a positive influence on protection motivation. Among the flood-affected, the sense of responsibility is generally high, see Table 5. Studies have shown that homeowners feel a greater sense of responsibility (Dillenardt et al., 2022; Grothmann & Reusswig, 2006). As over 80 % of respondents were homeowners, see Table 2, this might explain the high sense of responsibility observed. At the same time, respondents also place responsibility on the public authorities. Thus, those two perceptions are not mutually exclusive. This is in the spirit of integrated flood risk management. However, over 70 % of all respondents have little or rather little confidence that the public sector will fulfil the responsibilities they ascribe to it, see topic No. 11 in Table 5. This suggests that clear communication and confidence-building actions among all stakeholders involved in integrated flood risk management should be strengthened in the future.

**4.3 Framing factors: A chance to enhance adaptive behaviour?**

Framing factors offer the opportunity to discuss the influences of e.g. respondents' age, the availability of general or local information, the perceived availability of financial aid, and flood experience on adaptive behavior, i.e. the implementation of PLFRAM. The influence of framing factors is either indirect via the influence on threat, coping or responsibility appraisals, or direct, if the framing factor prevents the implementation of measures despite a high motivation of those affected and thus acts as a barrier. This study focuses on the indirect effects of the framing factors mentioned. The importance of framing factors for developing protective behavior was already addressed by Prentice-Dunn and Rogers (1986) in the Protection Motivation Theory, in which the influence of "source information" on threat and coping appraisal is mentioned. Lindell and Perry (2012) extend this understanding by stating that those factors form a framework, i.e. they are both at the beginning of the development of a protective response (indirect influence), i.e. they can directly hinder or promote the implementation of protection motivation in a protective response. Although the naming of this group of factors differs, other studies discuss framing factors. Fuchs et al. (2017) describe "situational factors", which include "being informed", for example, and assign them to a superclass of "social capital", which is assumed to have a positive influence on the implementation of measures.

The regression analysis of the framing factors shows low R-squared values. This is a known problem in psychological research. It is due to the fact that people are very different, but they do not participate in interviews that last longer than 30 minutes, making it impossible to include all personal and contextual factors (Grothmann & Reusswig, 2006). However,

conclusions from the results should be drawn with caution. Our analyzes show that home ownership indirectly promotes the motivation to protect oneself by strengthening coping and responsibility appraisals, which is in line with Grothmann and Reusswig (2006), who showed that ownership as a framing factor can positively influence the implementation of measures.

We found in the context of fluvial and flash flooding that the age of affected respondents negatively influenced their self-efficacy. Hence, older people, if they have experienced rather severe flooding, are less likely to see themselves in a position to implement PLFRAM. However, Houston et al. (2021) found, that households with older adults show less long term flood impacts and suggests, that this is caused by their social capital (e.g. social networks, knowledge). Information campaigns should built up on this and pay particular attention to older people in flooded areas by, enhancing or, if possible, profiting of their social capital, but to the same time identifying who could help them during the implementation process of PLFRAM

and recommending that they not select measures that require action during an event, such as mobile devices that need to be installed. Both perceived response efficacy and perceived self-responsibility were identified as the clearest triggers of protection motivation in the regression analysis presented in Table 6. Since they are enhanced by the perceived availability of financial aid, communicating financial aid may be crucial to support the implementation of adaptive measures. This

argument is strengthened by the findings of Houston et al. (2021) who shows a sensitivity to individuals' vulnerability and resilience to financial resources.

Our data suggests that those with little flood experience, i.e. those affected by urban pluvial or flash floods, were particularly likely to take action after the last flood. In contrast, those with more flood experience, i.e., those affected by fluvial floods, were particularly likely to have taken action before the last flood and were less likely to take further PLFRAM. In this

context, however, it should be considered that the flood experience is not only characterized by the pure experience of the flood, but also by the experience of the reconstruction process and possibly a subsequent adapted integrated flood risk management, as was the case, for example, after the 2002 and 2013 floods in Saxony (Müller, 2013). Such management, which includes the creation of flood hazard maps and information campaigns aimed at the population, may have a beneficial effect on peoples' perceptions of the threat, coping options, and responsibilities. Past research showed a positive effect of

(targeted) information campaigns on flood adaptation (Erdlenbruch & Bonté, 2018). While we cannot examine this relationship based on our data, we do find that those who have been affected by fluvial floods – who according to our data are also those who have the most flood experience – have a higher risk perception, a higher perceived response efficacy, a higher sense of personal responsibility, and a higher motivation to protect themselves, and feel better informed by their communities, see Table 5, and were more likely to have had implemented PLFRAM before the last flood event, see Figure 4.

Future research should focus on these relationships in order to better understand the extent to which integrated flood risk management of fluvial floods has had a positive impact on the adaptive behaviors of households. In the context of different types of flooding, it should then be considered whether similar management approaches should be adapted and applied to other types of flooding.

**4.4 Protection motivation and emotional coping: an interaction still not sufficiently understood**

Overall, the protection motivation of all respondents is positive or rather positive, and especially those who were affected by fluvial flooding have a high motivation to protect themselves from future events. At the same time, most of the interviewees agree with statements that indicate they will face future events with denial and fatalism. Denial and fatalism are markers of a non-protective response as defined by Grothmann and Reusswig (2006) and which is also referred to as emotional coping or maladaptive thinking in other studies. Grothmann and Reusswig (2006) conclude from their own and other studies that a

non-protective response has a negative/hindering effect on protection motivation.

Our results show for respondents who were affected by fluvial flooding that high ratings for denial and fatalism and a high protection motivation are not mutually exclusive but can instead coexist, which might be caused by repeated flooding and decreasing resilience, as indicated by other studies (Houston et al., 2021; Köhler et al., 2023). This may indicate that if the assessments of threat, coping, and (personal) responsibility are high, a protective motivation is promoted regardless of

maladaptive thinking. However, we found that those affected by fluvial flooding implemented fewer measures after the event than the other respondents. This might be a hint that a protective response is the result of the interaction between maladaptive thinking and protection motivation. Our data show that, at least in the context of fluvial flooding, the high sense of self-responsibility is not enough to hinder those affected from developing a non-protective response, although self-responsibility was found to have a hindering effect in on maladaptive thinking in the context of urban pluvial flooding

(Dillenardt et al., 2022). Hence, interconnections among the factors of PMT and PADM are not yet fully understood, in particular in the context of different flood types, and the exact role of maladaptive thinking cannot be conclusively clarified. Further research is needed on this topic. For instance, future research could use qualitative interviews to identify or confirm items to capture maladaptive thinking within future survey campaigns.

**4.5 Limitations**

In this study, we compare people affected by different types of flooding. Therefore, we conducted several surveys. Between 2013 and today, our survey methodology has evolved away from computer-aided telephone interviews (CATI) towards comuter-assisted web interviews (CAWI), see Figure 2. The reason for this is that the use of mobile phones has increased

rapidly in the last decade, and it can no longer be assumed that a balanced sample can be reached via landline. As a result, the "fluvial" group is homogeneous in terms of methodology (CATI), while the "urban pluvial" and "flash" flooding groups are mixed in terms of sampling methods used. One thing to admit in this context is that it is hardly possible to derive response rates for a CAWI if it was advertised via social media, as it is impossible to conclusively determine how many people were reached by the advertising or the sharing of the survey link by those who were reached by the advertising. In

addition, a study conducted in Australia by Gilligan et al. (2014) indicates that participants recruited through Facebook may be more socially engaged, better educated and have higher earnings than the general population. In our study, however, the

CAWIs within a flood type group were not advertised exclusively via social media but also via direct mail. We assume that the mixed use of methods minimises those effects.

In addition to these limitations, which can be attributed to the mixture of sampling methods, it is possible that our surveys were unable to reach those affected who had moved to a new place of residence after experiencing flooding. This is supported by the fact that we received around 1/5 of the letters sent out by the municipality as part of the survey conducted in the wake of the 2021 flood as undeliverable. This group could, therefore, be underrepresented in Sampling D, see Figure 2. However, the applied mixture of sampling methods will likely reduce that effect within the overall group affected by flash flooding. Shaver et al. (2019) point out that Facebook uses a non-random targeting algorithm. Furthermore, our survey targeted exclusively affected households. Our sampling based on advertisement via Meta is, therefore, non-random, and our results only reflect the perceptions of those affected and not the perceptions of unaffected households. In addition, our surveys were conducted exclusively in Germany. The transfer to other regions must, therefore, be scrutinised in advance. For example, it can be assumed that the sense of responsibility of those affected by floods differs between different countries (Andrasko, 2021).

With regard to the PLFRAM implemented, this study and the available data cannot clarify the extent to which households adapted appropriately before or after the flood. This is because which PLFRAM or combinations of PLFRAM are appropriate to the individual flood risk depends on many individual and local factors for which no data was collected. Furthermore, it is not possible to conclusively clarify how much financial, time and/or construction effort was required by those affected to implement PLFRAM. This is because the classes used differentiate between PLFRAM in terms of their mode of action and not in terms of implementation costs or effort."

## 5 Conclusion

We examined and compared adaptive behavior of households that experienced urban pluvial flooding between 2014 and 2019, flash flooding in 2016 or 2021 or fluvial flooding in 2013 in Germany. Our findings are based on several post-event surveys that were analyzed descriptively, via Kruskal-Wallis tests, single-factor ANOVA and linear regressions. We used the theoretical frameworks of PMT and PADM to structure our analyzes and discuss our results in a way that allows us to draw practical conclusions for future risk communication strategies.

The communication of the threat should include the probability of future events in particular to those at risk of urban pluvial flooding, and communicate high flow velocities and water inundations if those are expected. The local context must be established so that those affected can become aware of their individual risk. Our results suggest that informing affected individuals about PLFRAM and responsibilities should be a focus of information campaigns. Flood type-specific recommendations and cost-benefit analyzes should be carried out. The results of such analyzes should be communicated to specific target groups so that the measures are adapted to the expected severity and hydraulic forces. Care should be taken to

ensure that the communicated PLFRAM can be implemented by the respective target group, e.g., evasion measures by those involved in house construction. It may be advisable to incorporate the implementation of PLFRAM into the planning and permitting process. As respondents show very little trust in the public sector with regard to dealing with floods, especially after events that are perceived as very severe, communication strategies should include confidence-building strategies and

clearly communicate responsibilities. Particularly after a flood event, those affected are open to information campaigns, but those campaigns should be flood type-specific.

Our results suggest that investigating framing factors enhances the discussion about households' adaptive behaviour. In this context, we discussed weather and how the perceived availability of information and financial aid, flood experience, and homeownership promotes aspects of PMT and PADM. We found that the perceived availability of financial aid and

information positively impacts coping appraisal and that community-led information campaigns are more likely to increase peoples' sense of personal responsibility. However, the interaction of these factors as well as the effect of maladaptive thinking within the development of an adaptive behavior is not yet sufficiently understood, neither in our study nor in the wider literature. Further research is needed, as a better understanding can strengthen future risk communication strategies.

**Competing interests**

The contact author has declared that none of the authors has any competing interests.

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

| | Description | Item in survey(-s) | Survey(-s) in which the item was asked |
|---|---|---|---|
| **Evasion** | Measures that remove the entire building out of the risk zone. | Moving to a less threatened area. | S-3; S-1; S-6 |
| | | Upstands (e.g. steps) | S-4; S-5; S-2; S-6; |
| | | Dispensing with a cellar | S-2; S-3; S-1; S-6 |
| | | Driveways dropping towards the street | S-4; S-5 |
| **Resistance** | Measures that do not allow the water to enter the building when it reaches the building. | Ground sills | S-4; S-5 |
| | | Barrier systems or safety gates | S-4; S-5 |
| | | Backflow flap | S-4; S-5; S-2; S-3; S-1; S-6 |
| | | Waterproof or pressure-resistant windows and/or doors | S-4; S-5 |
| | | Window flaps or stationary or mobile water stops | S-4; S-5; S-2; S-3; S-1; S-6 |
| | | Waterproofing of the foundation | S-4; S-5 |
| | | Improvement of the flood safety of the building, e.g. improved structural stability | S-2; S-3 |
| **Drawback** | Measures that reduce loss caused by water penetration. Measures that reduce loss due to the protection of pollutants are excluded, as these are listed in a separate category "Securing". | I improve the flood safety of my building, i.e. I improve the stability of the building | S-1; S-6 |
| | | Low-value use of the floors at risk of flooding | S-4; S-5; S-2; S-3; S-1; S-6 |
| | | Low-value use of the floors at risk of flooding | S-4; S-5; S-2; S-3; S-1; S-6 |
| | | Buying pumps | S-4; S-5; S-2; S-3; S-1; S-6 |
| **Securing** | Measures that reduce loss from floodwater intrusion by protecting hazardous materials and pollutants. | Heating oil protection or relocation of the heating system and/or electrical utilities to higher floors | S-4; S-5 |
| | | Relocation of the heating system and/or the electrical utilities to higher floors | S-3; S-2; S-1; S-6 |
| | | Not storing varnish, paint or gasoline cans in the basement | S-2; S-3; S-1; S-6 |
| | | Changing the heating system or providing the oil tank with flood protection | S-2; S-3; S-1; S-6 |
| **Risk transfer** | Measures that do not directly prevent loss from flooding but transfer the cost of the loss to someone else. | Insurance against flood loss | S-4; S-5; S-2; S-3; S-1; S-6 |
| **Behaviour precaution** | Measures that cannot be implemented because they are changes in behaviours or the acquisition of new behaviours. Here we also include information seeking, as this can be considered a protective behaviour (Maidl & Buchecker, 2015). | Preparations for the eventuality of a hazard | S-4; S-5; S-2; S-3; S-1; S-6 |
| | | Search for information on how affected individuals can protect themselves | S-2; S-3; S-1; S-6 |
| | | Participation in seminars | S-2; S-3; S-6 |
| | | Participation in neighborhood networks | S-2; S-3; S-1; S-6 |
| | | Informing oneself about one's risk | S-1; S-6 |
| | | Acquisition of an emergency generator or a power generator | S-2; S-3; S-1; S-6 |

**Table B: Items asked in the surveys, their respective scale, and the respondents' answers in percent.**

| No. | topic | Item asked in survey | Flood type | Scale | 1 | 2 | 3 | 4 | 5 | 6 |
|---|---|---|---|---|---|---|---|---|---|---|
| | | | | Values represent the proportion of respondents [%] who rated the respective item with the respective number. | | | | | | |
| 1 | Flow velocity | What best describes the water velocity? | Fluvial | 1- steady flow 6- turbulent flow | 46 | 11 | 14 | 11 | 7 | 10 |
| 2 | | | Pluvial | | 30 | 14 | 17 | 13 | 10 | 15 |
| 3 | | | Flash | | 6 | 8 | 12 | 20 | 21 | 32 |
| 4 | Flow velocity and inundation | An average man… | Fluvial | 1: …could have stood with no difficulty 2: …could have stood only with difficulty 3: …would have been swept away. 4: Water too deep to stand. | 59 | 20 | 18 | 3 | --- | --- |
| 5 | | | Pluvial | | 75 | 14 | 11 | 0 | --- | --- |
| 6 | | | Flash | | 22 | 22 | 36 | 20 | --- | --- |
| 7 | Perceived probability | How likely do you think it is that your apartment or house will be hit by flooding again? | Fluvial | 1 - Very unlikely 6 - Very likely | 5 | 8 | 16 | 11 | 17 | 44 |
| 8 | | | Pluvial | | 13 | 16 | 27 | 17 | 14 | 13 |
| 9 | | | Flash | | 6 | 14 | 25 | 24 | 14 | 18 |
| 10 | Perceived severity | How bad do you expect the consequences of a future event to be? | Fluvial | 1 - Very bad 6 - Not bad at all | 38 | 20 | 21 | 8 | 7 | 7 |
| 11 | | | Pluvial | | 14 | 17 | 31 | 17 | 11 | 9 |
| 12 | | | Flash | | 34 | 22 | 20 | 14 | 7 | 3 |
| 13 | Response efficacy | Adaptive measures can significantly reduce flood damage. | Fluvial | 1 - I fully agree 6 - I do not agree at | 40 | 16 | 19 | 6 | 6 | 13 |
| 14 | | | Pluvial | | 37 | 20 | 21 | 7 | 5 | 10 |
| 15 | | | Flash | | 24 | 16 | 18 | 17 | 14 | 10 |
| 16 | Response cost | Adaptive measures are far too expensive. | Fluvial | 1 - I fully agree 6 - I do not agree at | 26 | 20 | 24 | 10 | 7 | 12 |
| 17 | | | Pluvial | | 16 | 18 | 27 | 12 | 11 | 17 |
| 18 | | | Flash | | 21 | 14 | 22 | 16 | 12 | 16 |
| 19 | Self-efficacy | Personally, I do not feel able to implement any of the measures mentioned above. | Fluvial | 1 - I fully agree 6 - I do not agree at | 11 | 8 | 16 | 8 | 15 | 42 |
| 20 | | | Pluvial | | 9 | 9 | 19 | 9 | 14 | 40 |
| 21 | | | Flash | | 8 | 10 | 12 | 15 | 13 | 42 |
| 22 | Responsibility public | Flood prevention is the responsibility of public institutions and not of private individuals. | Fluvial | 1 - I fully agree 6 - I do not agree at | 23 | 15 | 34 | 8 | 7 | 13 |
| 23 | | | Pluvial | | 17 | 17 | 31 | 12 | 10 | 12 |
| 24 | | | Flash | | 21 | 16 | 25 | 18 | 8 | 12 |
| 25 | Responsibility self | Every individual has a responsibility to reduce flood damage as much as possible. | Fluvial | 1 - I fully agree 6 - I do not agree at | 61 | 21 | 12 | 2 | 2 | 3 |
| 26 | | | Pluvial | | 39 | 25 | 20 | 6 | 5 | 5 |
| 27 | | | Flash | | 29 | 22 | 19 | 14 | 8 | 9 |

| # | Category | Statement | Flood type | Scale | 1 | 2 | 3 | 4 | 5 | 6 |
|---|---|---|---|---|---|---|---|---|---|---|
| 28 | Fatalism | There is generally nothing that can be done about flooding and flood damage. | Fluvial | 1 - I fully agree | 31 | 13 | 23 | 9 | 10 | 14 |
| 29 | | | Pluvial | 6 - I do not agree at | 17 | 15 | 22 | 12 | 14 | 20 |
| 30 | | | Flash | | 15 | 13 | 15 | 20 | 17 | 20 |
| 40 | Denial | I don't even want to think about future flood damage! | Fluvial | 1 - I fully agree | 72 | 8 | 9 | 3 | 2 | 6 |
| 41 | | | Pluvial | 6 - I do not agree at | 36 | 14 | 19 | 8 | 8 | 14 |
| 42 | | | Flash | | 25 | 18 | 17 | 13 | 10 | 17 |
| 43 | Trust | The flood protection in our region is so good that I don't need to take private protection measures. | Fluvial | 1 - I fully agree | 3 | 4 | 9 | 9 | 20 | 55 |
| 44 | | | Pluvial | 6 - I do not agree at | 9 | 9 | 16 | 12 | 18 | 36 |
| 45 | | | Flash | | 2 | 2 | 5 | 7 | 17 | 68 |
| 46 | | There are enough tax concessions and subsidy programs for financing adaptive measures. | Fluvial | 1 - I fully agree | 8 | 8 | 19 | 12 | 19 | 35 |
| 47 | | | Pluvial | 6 - I do not agree at | 5 | 9 | 20 | 17 | 19 | 30 |
| 48 | | | Flash | | 5 | 4 | 7 | 14 | 26 | 43 |
| 49 | Information available | There is far too little information and advice on private flood prevention. | Fluvial | 1 - I fully agree | 21 | 14 | 21 | 10 | 13 | 21 |
| 50 | | | Pluvial | 6 - I do not agree at | 17 | 20 | 24 | 11 | 11 | 18 |
| 51 | | | Flash | | 21 | 16 | 21 | 17 | 12 | 13 |
| 52 | | Our municipality provides very good information about flood risks and possible precautionary measures. | Fluvial | 1 - I fully agree | 26 | 19 | 19 | 9 | 10 | 18 |
| 53 | | | Pluvial | 6 - I do not agree at | 9 | 12 | 17 | 13 | 17 | 32 |
| 54 | | | Flash | | 8 | 6 | 12 | 14 | 21 | 39 |
| 55 | Protection motivation | Personally, I will do everything I can to protect the house I live in from flooding. | Fluvial | 1 - I fully agree | 76 | 11 | 5 | 2 | 1 | 5 |
| 56 | | | Pluvial | 6 - I do not agree at all | 48 | 24 | 15 | 6 | 3 | 5 |
| 57 | | | Flash | | 31 | 19 | 21 | 16 | 7 | 6 |

740