# Peer review of "Individual Flood Risk Adaptation in Germany: Exploring the Role of Different Types of Flooding"

_EGUsphere, 2024_

## Author Comment (AC2)

| Reviewer 1 | | |
|---|---|---|
| No. | Comment | Answer |
| | This paper investigates the influence of different types of flooding on adaptive behavior and risk communication in Germany. The authors use survey data from over 3000 households affected by fluvial, flash, and urban pluvial floods to examine the factors that influence adaptive behavior and the effectiveness of different types of adaptive measures. The findings suggest that there are flood type-specific differences in adaptive responses, with fluvial flood-affected households implementing measures before the event but showing signs of emotional coping, while flash flood-affected households are more likely to implement measures after the event. However, the lack of detailed methodology and comparisons with existing literature limit the paper's overall quality. This paper still needs a major revision before it could be acceptable for publication. | Thank you for reviewing our manuscript. Your comments will help us improve the paper. Please find below a point-by-point response how we are going to revise the manuscript. |
| 1 | The paper lacks a detailed description of how to collect and analyze the survey data. Authors should provide more details on the methodology section. Specifically, how was the sample selected, and what statistical techniques were used to analyze the data? It would be useful to provide more information on the survey design, sampling methods, and data analysis techniques to help the readers. | To clarify our sampling methodology, we will move the paragraph on this to the beginning of Chapter 2, "Data & Methods.". The Chapter starts as follows in the revised version of the manuscript:

"This study is based on survey data collected via four different survey designs (see figure 2) between 2014 and 2022 in the course of six surveys among flood-affected households in Germany (see Table 1). While S-1, S-2, S-3, and S-4 were created by a random sampling in affected areas (based on lists of flooded roads; see Thieken et al. 2017) and considered only landlines, S6 was created in Rhineland-Palatinate with the help of the district Ahrweiler, where every third household who had applied for |

immediate disaster aid was invited to participate. In North Rhine-Westphalia (as well as in S-5) people from the affected areas were invited to an online survey via advertisements on Facebook and other media. Advertising via Meta to recruit survey participants is a method used in health-related research during the last decades (Gilligan et al., 2014; Kapp et al., 2013; Shaver et al., 2019), A total of 3670 households were questioned about the impacts of recently experienced flood events along with questions on adaptive behaviour based on concepts from the PMT and PADM. Data were collected by paper/pencil, computer-assisted web interview (CAWI), and/or computer-assisted telephone interviews (CATI), see table 1."

To explain the sampling in more detail, we will create a new figure (as Figure 2) that provides an overview of the sampling methods. In addition, the samples in Table 1 will be linked to the new Figure 2.

Please bot that we already explained the data analyses in the paper; to enhance clarity, we will update the text as follows:

„We analysed the data using the statistical software package IBM SPSS 27. To identify significant differences between the three flood types, the Kruskal-Wallis test was performed. For each PMT factor, a Kruskal-Wallis test was first performed with all three flood types. If the Kruskal-Wallis test showed that there was no significant difference between the flood types, this was indicated in Table 4 and no post-hoc test was performed. If the Kruskal-Wallis test showed significant differences, single-factor ANOVAs were performed to better understand identified differences by comparing the flood types in pairs.

Linear regressions were carried out with IBM SPSS 27 to examine in the first step which PMT/PADM factors, i.e., threat, coping and responsibility appraisal, influenced the protection motivation of the respondents. The dependent variable for the regressions presented in table 6 was protection motivation, which we derived from the items "I will do everything possible to protect myself from flooding" and the item "I would recommend that others take private

| | | precautions" (see Table B1). These two items were combined so that the highest value was always taken for the combined variable. This combined variable enables us to capture protection motivation regardless of whether it relates to the respondent, as in the first item, or to others, as in the second item. In a second step, the PMT/PADM factors that significantly influenced protection motivation were examined to determine the framing factors that influenced them." |
|---|---|---|
| 2 | The paper could benefit from a more in-depth discussion of the limitations of the study, such as the potential biases in the survey data and the generalizability of the findings to other regions. For example, have you considered the potential biases in the survey data, such as non-response bias or selection bias? How do these biases affect the generalizability of your findings? | We will include a sub-chapter entitled "Limitations" at the end of Chapter 4, in which we discuss this work's limitations as follows:

"In this study, we compare people affected by different types of flooding between 2013 and 2021 based on several surveys. Over the years, the survey methodology has evolved away from CATI towards CAWI. Due to the rapidly increasing use of mobile phones it can no longer be assumed that a balanced sample can be reached via landlines that are used in CATI. In fact, younger people tend to become underrepresented in CATIs. Therefore, these were accompanied or entirely substituted by CAWI. As a result, the "fluvial" group is homogeneous in terms of methodology (CATI), while the "urban pluvial" and "flash" flooding groups are mixed in terms of sampling methods used. While age groups are now better represented in CAWIs, it is hardly possible to derive response rates for a CAWI if it was advertised via social media, as it is impossible to conclusively determine how many people were reached by the advertising or the sharing of the survey link by those who were reached by the advertising. In addition, a study conducted in Australia by Gilligan et al. (2014) indicates that participants recruited through Facebook may be more socially engaged, better educated and have higher earnings than the general population. In our study, however, the CAWIs within a flood-type group were not advertised exclusively via social media but also via direct mail (i.e., in the district of Ahrweiler) or advertisements and reports in local newspapers. We assume that the mixed use of methods minimises those effects.

Our survey targeted exclusively affected |

| | | households. Therefore, our results only reflect the perceptions of those affected and not the perceptions of unaffected households. Shaver et al. (2019) point out that Facebook uses a non-random targeting algorithm. In addition, our surveys were conducted exclusively in Germany. The transfer to other regions must, therefore, be scrutinised in advance. For example, it can be assumed that the sense of responsibility of those affected by floods differs between different countries (Andrasko, 2021). Therefore, one aim of future research should be to collect data continuously and across national borders to investigate the transferability of our and other study results regarding individuals' adaptation and adaptive behaviour." |
| | | |
| | | With regard to the PLFRAM implemented, this study and the available data cannot clarify the extent to which households adapted appropriately before or after the flood. This is because which PLFRAM or combinations of PLFRAM are appropriate to the individual flood risk depends on many individual and local factors for which no data was collected. Furthermore, it is not possible to conclusively clarify how much financial, time and/or construction effort was required by those affected to implement PLFRAM. This is because the classes used differentiate between PLFRAM in terms of their mode of action and not in terms of implementation costs or effort." |
| 3 | The paper would be strengthened by including comparisons with other related research in the field of flood risk adaptation to provide a more comprehensive evaluation of the conclusion. I think it is also necessary to compare your findings with existing literature on flood risk adaptation. It would be valuable to discuss how your results align with or differ from previous studies in the field. | In our discussion, we would suggest the following additional comparisons and references to other studies and research in the field of risk adaptation:

 - The importance of framing factors for developing protective behaviour has already been recognised in other studies, although the naming of this group of factors differs. Fuchs et al. (2017) describe "situational factors", which include "being informed", for example, and assign them to a superclass of "social capital", which is assumed to have a positive influence on the implementation of measures. Grothmann and Reusswig (2006) speak of personal or contextual factors potentially influencing people's behaviour. Bubeck et al. (2018) distinguish between environmental and intrapersonal factors |

| | | | influencing threat and coping appraisal. |
|---|---|---|---|
| | | | - The regression analysis in table 6, reveals no significant link between perceived probability of a future event and protection motivation for fluvial and flash flooding, what is in line with findings in Australia (Bird et al., 2013). |
| | | | - The regression analysis of the framing factors shows low R-squared values. This is a known problem in psychological research. It is due to the fact that people are very different, but they do not participate in surveys that last longer than 30 minutes, making it impossible to include all personal and contextual factors (Grothmann & Reusswig, 2006). |
| | | | - Our analyses show that home ownership indirectly promotes the motivation to protect oneself by strengthening coping and responsibility appraisals, which is in line with Grothmann and Reusswig (2006), who showed that ownership as a framing factor can positively influence the implementation of measures. |
| | | | - Hence, older people, if they have experienced rather severe flooding, are less likely to see themselves in a position to implement measures. Brockie and Miller (2017) found that older adults rely on social capital during and after flooding. However, Houston et al. (2021) found, that households with older adults even show less long term flood impacts and suggest that it is this is caused by social capital (e.g. social networks, knowledge). |
| | | | - Since perceived response efficacy and perceived self-responsibility are enhanced by the perceived availability of financial aid, communicating financial aid may be crucial to support the implementation of adaptive measures. This argument is strengthened by the fact that Houston et al. (2021) show a sensitivity to individuals' vulnerability and resilience to financial resources. |
| | | | - Past research showed a positive effect of (targeted) information campaigns on flood adaptation (Erdlenbruch & Bonté, 2018). |
| | | | - In North Rhine-Westphalia (as well as in S-5) people from the affected areas were invited for a CAWI via advertisements on |

| | | Facebook and other media. Advertising via Meta to recruit survey participants is a method used in health-related research during the last decades (Gilligan et al., 2014; Kapp et al., 2013; Shaver et al., 2019). Thieken et al. (2023) advertised a survey via Meta and "did not find any anomalies concerning the age distribution of the respondents in the data collected in this way. |
|---|---|---|
| 4 | Besides, the format of this manuscript is poor, especially the placement of the text in the tables, and the images have the low resolution. These problems need to be carefully resolved. | We will revise both the figures and the tables. |

**References**

Andrasko, I. (2021). Why People (Do Not) Adopt the Private Precautionary and Mitigation Measures: A Review of the Issue from the Perspective of Recent Flood Risk Research. *Water*, *13*(2). https://doi.org/ARTN 14010.3390/w13020140

Bird, D., King, D., Haynes, K., Box, P., Okada, T., & Nairn, K. (2013). *Impact of the 2010–11 floods and the factors that inhibit and enable household adaptation strategies*. https://nccarf.edu.au/wp-content/uploads/2019/03/Bird_2013_Floods_household_adaptation_strategies.pdf

Brockie, L., & Miller, E. (2017). Understanding Older Adults' Resilience During the Brisbane Floods: Social Capital, Life Experience, and Optimism. *Disaster Medicine and Public Health Preparedness*, *11*(1), 72-79. https://doi.org/10.1017/dmp.2016.161

Bubeck, P., Wouter Botzen, W. J., Laudan, J., Aerts, J. C. J. H., & Thieken, A. H. (2018). Insights into Flood-Coping Appraisals of Protection Motivation Theory: Empirical Evidence from Germany and France. *Risk Analysis*, *38*(6), 1239-1257. https://doi.org/10.1111/risa.12938

Erdlenbruch, K., & Bonté, B. (2018). Simulating the dynamics of individual adaptation to floods. *Environmental Science & Policy*, *84*, 134-148. https://doi.org/https://doi.org/10.1016/j.envsci.2018.03.005

Fuchs, S., Karagiorgos, K., Kitikidou, K., Maris, F., Paparrizos, S., & Thaler, T. (2017). Flood risk perception and adaptation capacity: a contribution to the socio-hydrology debate. *Hydrol. Earth Syst. Sci.*, *21*(6), 3183-3198. https://doi.org/10.5194/hess-21-3183-2017

Gilligan, C., Kypri, K., & Bourke, J. (2014). Social Networking Versus Facebook Advertising to Recruit Survey Respondents: A Quasi-Experimental Study [Original Paper %J JMIR Res Protoc]. *3*(3), e48. https://doi.org/10.2196/resprot.3317

Grothmann, T., & Reusswig, F. (2006). People at Risk of Flooding: Why Some Residents Take Precautionary Action While Others Do Not. *Natural Hazards*, *38*(1), 101-120. https://doi.org/10.1007/s11069-005-8604-6

Houston, D., Werritty, A., Ball, T., & Black, A. (2021). Environmental vulnerability and resilience: Social differentiation in short- and long-term flood impacts. *46*(1), 102-119. https://doi.org/https://doi.org/10.1111/tran.12408

Kapp, J. M., Peters, C., & Oliver, D. P. (2013). Research Recruitment Using Facebook Advertising: Big Potential, Big Challenges. *Journal of Cancer Education*, *28*(1), 134-137. https://doi.org/10.1007/s13187-012-0443-z

Shaver, L. G., Khawer, A., Yi, Y., Aubrey-Bassler, K., Etchegary, H., Roebothan, B., Asghari, S., & Wang, P. P. (2019). Using Facebook Advertising to Recruit Representative Samples: Feasibility Assessment of a Cross-Sectional Survey [Original Paper %J J Med Internet Res]. *21*(8), e14021. https://doi.org/10.2196/14021

Thieken, A. H., Bubeck, P., Heidenreich, A., von Keyserlingk, J., Dillenardt, L., & Otto, A. (2023). Performance of the flood warning system in Germany in July 2021 – insights from affected residents. *Nat. Hazards Earth Syst. Sci.*, *23*(2), 973-990. https://doi.org/10.5194/nhess-23-973-2023

---

## Author Response (AR1)

Dear Marvin Ravan, and Reviewers,

On behalf of all the authors, I would like to thank you for the constructive comments and criticism received on our manuscript entitled 'Individual Flood Risk Adaptation in Germany: Exploring the Role of Different Types of Flooding'. We believe that in the current revision we have addressed the comments raised by the reviews and that in doing so our revised manuscript is now more suitable for publication in Natural Hazards and Earth System Sciences.

We are looking forward to your comments.

| | |
|---|---|
| Dear Marvin Ravan,

In addition to the changes made in response to the reviewers' comments, we have made the following changes to the manuscript to improve its quality: | |
| No. | Description |
| 1 | We have spellchecked the text again. This has resulted in (a) corrections to the grammar and (b) standardised spelling of technical terms used. |
| 2 | We made a typing error in the description of the data, which we have corrected. On page 5, line 146, the number of excluded cases in Remscheid was corrected from 64 to 53. |
| 3 | Following the remarks from the preceding review file validation, we have removed the colouring of the cells in Table 7 on page 17. |

| Reviewer 1 | | |
|---|---|---|
| No. | Comment | Answer |
| | This paper investigates the influence of different types of flooding on adaptive behavior and risk communication in Germany. The authors use survey data from over 3000 households affected by fluvial, flash, and urban pluvial floods to examine the factors that influence adaptive behavior and the effectiveness of different types of adaptive measures. The findings suggest that there are flood type-specific differences in adaptive responses, with fluvial flood-affected households implementing measures before the event but showing signs of emotional coping, while flash flood-affected | Thank you for reviewing our manuscript. Your comments will help us improve the paper. Please find below a point-by-point response how we revised the manuscript. |

| | | households are more likely to implement measures after the event. However, the lack of detailed methodology and comparisons with existing literature limit the paper's overall quality. This paper still needs a major revision before it could be acceptable for publication. | |
|---|---|---|---|
| 1 | | The paper lacks a detailed description of how to collect and analyze the survey data. Authors should provide more details on the methodology section. Specifically, how was the sample selected, and what statistical techniques were used to analyze the data? It would be useful to provide more information on the survey design, sampling methods, and data analysis techniques to help the readers. | To clarify our sampling methodology, moved the paragraph on this to the beginning of Chapter 2, "Data & Methods.". The Chapter starts as follows in the revised version of the manuscript:

"This study is based on survey data collected via four different survey designs (see figure 2) between 2014 and 2022 in the course of six surveys among flood-affected households in Germany (see Table 1). While S-1, S-2, S-3, and S-4 were created by a random sampling in affected areas (based on lists of flooded roads; see Thieken et al. 2017) and considered only landlines, S6 was created in Rhineland-Palatinate with the help of the district Ahrweiler, where every third household who had applied for immediate disaster aid was invited to participate. In North Rhine-Westphalia (as well as in S-5) people from the affected areas were invited to an online survey via advertisements on Facebook and other media. Advertising via Meta to recruit survey participants is a method used in health-related research during the last decades (Gilligan et al., 2014; Kapp et al., 2013; Shaver et al., 2019). A total of 3670 households were questioned about the impacts of recently experienced flood events along with questions on adaptive behaviour based on concepts from the PMT and PADM. Data were collected by paper/pencil, computer-assisted web interview (CAWI), and/or computer-assisted telephone interviews (CATI), see table 1." (see page 4, line 116ff)

To explain the sampling in more detail, we will create a new figure (see figure 2 at page 5) that provides an overview of the sampling methods. In addition, the samples in Table 1 on page 6 will be linked to the new Figure 2.

Please bot that we already explained the data |

| | | analyses in the paper; to enhance clarity, we will update the text as follows:

„We analysed the data using the statistical software package IBM SPSS 27. To identify significant differences between the three flood types, the Kruskal-Wallis test was performed. For each PMT factor, a Kruskal-Wallis test was first performed with all three flood types. If the Kruskal-Wallis test showed that there was no significant difference between the flood types, this was indicated in Table 4 and no post-hoc test was performed. If the Kruskal-Wallis test showed significant differences, single-factor ANOVAs were performed to better understand identified differences by comparing the flood types in pairs.

Linear regressions were carried out with IBM SPSS 27 to examine in the first step which PMT/PADM factors, i.e., threat, coping and responsibility appraisal, influenced the protection motivation of the respondents. The dependent variable for the regressions presented in table 6 was protection motivation, which we derived from the items "I will do everything possible to protect myself from flooding" and the item "I would recommend that others take private precautions" (see Table B1). These two items were combined so that the highest value was always taken for the combined variable. This combined variable enables us to capture protection motivation regardless of whether it relates to the respondent, as in the first item, or to others, as in the second item. In a second step, the PMT/PADM factors that significantly influenced protection motivation were examined to determine the framing factors that influenced them." (see page 7, line 174ff) |

| 2 | The paper could benefit from a more in-depth discussion of the limitations of the study, such as the potential biases in the survey data and the generalizability of the findings to other regions. For example, have you considered the potential biases in the survey data, such as non-response bias or selection bias? How do these biases affect the generalizability of your findings? | We included a sub-chapter entitled "Limitations" at the end of Chapter 4, in which we discuss this work's limitations as follows (see page 22, line 528ff):

"In this study, we compare people affected by different types of flooding between 2013 and 2021 based on several surveys. Over the years, the survey methodology has evolved away from CATI towards CAWI. Due to the rapidly increasing use of mobile phones it can no longer be assumed that a balanced sample can be reached via landlines that are used in CATI. In fact, |

younger people tend to become underrepresented in CATIs. Therefore, these were accompanied or entirely substituted by CAWI. As a result, the "fluvial" group is homogeneous in terms of methodology (CATI), while the "urban pluvial" and "flash" flooding groups are mixed in terms of sampling methods used. While age groups are now better represented in CAWIs, it is hardly possible to derive response rates for a CAWI if it was advertised via social media, as it is impossible to conclusively determine how many people were reached by the advertising or the sharing of the survey link by those who were reached by the advertising. In addition, a study conducted in Australia by Gilligan et al. (2014) indicates that participants recruited through Facebook may be more socially engaged, better educated and have higher earnings than the general population. In our study, however, the CAWIs within a flood-type group were not advertised exclusively via social media but also via direct mail (i.e., in the district of Ahrweiler) or advertisements and reports in local newspapers. We assume that the mixed use of methods minimises those effects.

Our survey targeted exclusively affected households. Therefore, our results only reflect the perceptions of those affected and not the perceptions of unaffected households. Shaver et al. (2019) point out that Facebook uses a non-random targeting algorithm. In addition, our surveys were conducted exclusively in Germany. The transfer to other regions must, therefore, be scrutinised in advance. For example, it can be assumed that the sense of responsibility of those affected by floods differs between different countries (Andrasko, 2021). Therefore, one aim of future research should be to collect data continuously and across national borders to investigate the transferability of our and other study results regarding individuals' adaptation and adaptive behaviour."

With regard to the PLFRAM implemented, this study and the available data cannot clarify the extent to which households adapted appropriately before or after the flood. This is because which PLFRAM or combinations of PLFRAM are appropriate to the individual flood risk depends

| | | on many individual and local factors for which no data was collected. Furthermore, it is not possible to conclusively clarify how much financial, time and/or construction effort was required by those affected to implement PLFRAM. This is because the classes used differentiate between PLFRAM in terms of their mode of action and not in terms of implementation costs or effort." |
|---|---|---|
| 3 | The paper would be strengthened by including comparisons with other related research in the field of flood risk adaptation to provide a more comprehensive evaluation of the conclusion. I think it is also necessary to compare your findings with existing literature on flood risk adaptation. It would be valuable to discuss how your results align with or differ from previous studies in the field. | In our discussion, we included the following additional comparisons and references to other studies and research in the field of risk adaptation:

- In North Rhine-Westphalia (as well as in S-5) people from the affected areas were invited for a CAWI via advertisements on Facebook and other media. Advertising via Meta to recruit survey participants is a method used in health-related research during the last decades (Gilligan et al., 2014; Kapp et al., 2013; Shaver et al., 2019). Thieken et al. (2023) advertised a survey via Meta and "did not find any anomalies concerning the age distribution of the respondents in the data collected in this way. (see page 5, line 120 ff)

- "The importance of framing factors for developing protective behavior was already addressed by Prentice-Dunn and Rogers (1986) in the Protection Motivation Theory, in which the influence of "source information" on threat and coping appraisal is mentioned. Lindell and Perry (2012) extend this understanding by stating that those factors form a framework, i.e. they are both at the beginning of the development of a protective response (indirect influence), i.e. they can directly hinder or promote the implementation of protection motivation in a protective response. Although the naming of this group of factors differs, other studies discuss framing factors. Fuchs et al. (2017) describe "situational factors", which include "being informed", for example, and assign them to a superclass of "social capital", which is assumed to have a positive influence on the implementation of measures." (see page 21, line 461 ff) |

| | | | - The regression analysis of PMT/PADM aspects, see Table 6, reveals no significant link between perceived probability of a future event and protection motivation for fluvial and flash flooding, what is in line with findings in Australia (Bird et al., 2013). (see page 19, line 419 ff)

- The regression analysis of the framing factors shows low R-squared values. This is a known problem in psychological research. It is due to the fact that people are very different, but they do not participate in interviews that last longer than 30 minutes, making it impossible to include all personal and contextual factors (Grothmann & Reusswig, 2006). (page 20, line 469 ff)

- Our analyses show that home ownership indirectly promotes the motivation to protect oneself by strengthening coping and responsibility appraisals, which is in line with Grothmann and Reusswig (2006), who showed that ownership as a framing factor can positively influence the implementation of measures. (see page 21, line 472 ff)

- Hence, older people, if they have experienced rather severe flooding, are less likely to see themselves in a position to implement measures. Brockie and Miller (2017) found that older adults rely on social capital during and after flooding. However, Houston et al. (2021) found, that households with older adults even show less long term flood impacts and suggest that it is this is caused by social capital (e.g. social networks, knowledge). (see page 21, line 476 ff)

- Since they are enhanced by the perceived availability of financial aid, communicating financial aid may be crucial to support the implementation of adaptive measures. This argument is strengthened by the findings of Houston et al. (2021) who shows a sensitivity to individuals' vulnerability and resilience to |

| | | financial resources.
(see page 21, line 483 ff)

- Past research showed a positive effect of (targeted) information campaigns on flood adaptation (Erdlenbruch & Bonté, 2018). (see page 21, line 494 f) |
|---|---|---|
| 4 | Besides, the format of this manuscript is poor, especially the placement of the text in the tables, and the images have the low resolution. These problems need to be carefully resolved. | We revised both the figures and the tables. |

| Reviewer 2 | | |
|---|---|---|
| No. | Comment | Answer |
| | The paper provides an overview on how experiencing a flooding event may impact people's attitudes towards these events. In particular, it does so by building on data collected among the German population that was exposed to three types of floods: fluvial floods, urban pluvial floods and flash floods (circa 3000 households). The analysis is framed within the framework of the Protection Motivation Theory (PMT) and the Protection Action Decision Model (PADM). The topics approached in the paper are extremely relevant as we can expect these events to become more and more common and more and more disruptive in a warming planet. The structure is adequate, but I am raising some points that I hope the authors would be happy to consider as a contribution to improve the quality of their manuscript: | Thank you for reviewing our manuscript. Your comments will help us improve the paper. Please find below a point-by-point response how we revised the manuscript. |
| 1 | It would be useful to quote the official documents (even though they may be in German) of the Federal Water Act mentioned on line 42/43. | We added the following text to the revised manuscript:

"In particular, floods that occur due to an overloaded drainage system can be excluded by member states when adhering to the plan. Germany made use of this option when adapting |

| | | the Federal Water Act (Wasserhaushaltsgesetz – WHG) in 2009 to the requirements of the Floods Directive (WHG, 2009). *Section 72 of the Federal Water Act defines flooding as "[...] a temporary inundation of land not normally covered by water, in particular by surface waters or by seawater entering coastal areas. This does not include flooding from sewage systems."* (see page 2, line 41 ff) |
|---|---|---|
| 2 | Regarding Figure 1, I understand it builds on previous papers that are, rightfully, cited, but where does the top sentence ("... are caused by the release of large quantities [...]") come from? Would not it be easier to have a full sentence? Are you providing a definition of floods? | Figure 1 shows the different definitions of the types of flooding discussed in this paper and their overlaps. For better readability, we have now formulated complete sentences. The definitions with the respective references can also be found in the text. (see Figure 1, page 3) |
| 3 | On line 89 you mention that "The PADM adds - among other variables - [...]". What does it add to? Compared to what? I guess, maybe, the PMT? | Yes, PADM adds to the PMT. We will change the text as follows:

"The PADM *adds to the basic constructs of the PMT* that individuals assess the extent to which they themselves (perceived self-responsibility) or public institutions (perceived government responsibility) are responsible for the implementation of measures and the idea of framing/context giving factors (Lindell & Perry, 2012)." (see page 4, line 89 ff) |
| 4 | On line 161 you mention "the average age". I would avoid using "average" as a term, as it may be understood in different ways according to the context. Are we talking about the mean? The median? | We use the term "MEAN age" throughout the revised manuscript and specified the text accordingly:
"The median age of the respondents was 59, which is approx. 8 years above the  *mean* age of the over 18s in the German population (DeStatis, 2014)" (see page 7, line 171 f) |
| 5 | I am wondering if the decision to reach out to people through Facebook (mentioned on line 166) may represent a cause of concerned over biased sample selection. Are not older people significantly less likely to be on social media? Also, is this a standard practice? If other studies approached respondents in the same way, it may be useful to say so. | Thank you for bringing this to our attention. Surveys have already been promoted via Facebook and social media in health-related studies. We refer to these applications in the revised text as follows:

"In North Rhine-Westphalia (as well as in S-5) people from the affected areas were invited for a CAWI via advertisements on Facebook and other media. *Advertising via Meta to recruit survey participants is a method used in health-related research during the last decades (Gilligan et al., 2014; Kapp et al., 2013; Shaver et al., 2019). Thieken et al. (2023) advertised a survey via Meta and "did not find any anomalies concerning the age distribution of the respondents in the data* |

| | | |
|---|---|---|
| | | *collected in this way."* (see page 5, line 120 ff)

However, we also want to point out that the groups studied, namely those affected by flash flooding and those affected by urban pluvial flooding, were not exclusively advertised in this way, as outlined in Table 1. The computer-assisted web interviews (CAWI) were advertised by writing invitation letters directly to those affected in 2021. For the surveys advertised in this way, there was always the option for those invited to receive the questionnaire in printed form so that we did not exclude people who were less Internet-savvy or without the opportunity to use the Internet from the surveys. Nevertheless, we agree that our sampling method may contain biases. We will discuss these possibilities in an additional chapter, "Limitations" (see page 23, line 525 ff), at the end of the paper as suggested in the other review and hope we will sufficiently address your concerns, too. |
| 6 | On Table 2, gender options are listed as "m/f/d". What does "d" stand for? | The "d" stands for "diverse". We explain this in the revised manuscript in the caption of Table 2. (see Table 2, page 8) |
| 7 | On the phrasing "Yet, most of those affected by flash, fluvial and urban pluvial foods [...]" (line 266) I am wondering if these words simply imply every respondent. Are not you interviewing people affected by these three types of floods? | Thanks for pointing this out. This is correct and we will change the text as follows:
"Yet, most of those affected by flooding (median: 1-2) have little confidence in public flood protection measures. Moreover, most people affected by flooding have little confidence in state financial aid (median: 2-3)." (see page 10, line 253 f) |
| 8 | The description of the statistics is clear but I am particularly concerned about one question and how it was measured (see Table 5). Every question (or most of them) measures the degree of agreement from 1 to 6, where 1 indicated full agreement. However, Question 1 seems to be reversed, where very low levels indicate a disagreement (not really a disagreement, but an expectation that the event may not manifest). I found this a rather confusing approach. In a sense, it could have been phrased as a statement like "Your apartment would be hit by flooding" and then a scale of agreement from 1 to 6 as all the others. I guess you could | Thank you for this clear and understandable comment. The questions and scales used in this paper correspond to those from our surveys which were initially created after the flood in 2002 and were expanded by PMT factors over the years. However, the item on perceived future probability (question 1) was consistently phrased in the presented way in all surveys. Of course, we cannot change the data anymore, but we will keep your comment in mind for future surveys.
However, to facilitate the interpretation of the data, we recoded some items in Table 5 and for the regressions so that for all items, low values represent a "low/decreasing effect on protection motivation according to PMT" of the respective statement. High values represent a "high/increasing effect on protection motivation according to PMT" of the respective statement. This resulted in changes to the "Results" and |

| | | |
|---|---|---|
| | revert the values and turn the measuring of this question into something closer to the others? (I hope this point is clear enough but I am more than happy to come back to it). | "Discussions" sections. |
| 9 | On Table 6, it would probably be easier for the reader to see the dependent variable pointed out in the table itself rather than in its description. At least, this is the standard approach in econometrics, where regression results are omnipresent. | We agree that the dependent variable is more easily accessible to the reader if it is mentioned directly in the tables. Therefore, we adapt Table 6 on page 14. |
| 10 | I have some points on the results of the regression as presented in Table 7. (a) It could be interesting to introduce event- fixed effects. Fiexed Effects models are straightforward to add in a simple Ordinary Least Square and would help capturing anything that is specific to that single event and that the other independent variables would not be able to capture, improving the fitness of the model. (b) Maybe test the errors for heteroskedasticity? This is one of the standard assumptions (see, for instance, Wooldridge's Introductory Econometrics) to guarantee consistent and unbiased estimates. If you were to find issues of heteroskedasticity, it could be useful to provide measures of robust standard errors. (c) I notice that you are also concerned by this in the pages that follow, but I was wondering if you could compare your R-squared to those from studies that adopted a similar approach. If the R-squared there are also found to be so small, a somehow less worrying issue should be raised for your single case (and maybe a methodological discussion for the whole field should be raised). Otherwise, if this low R-square is specific to your manuscript you may want to rethink your model. (d) One | According to (a): Fixed effects regression models fix variables that do not change over time, thereby removing their influence on the model. This method can only be applied to datasets that record how variables change over time, so it can only be applied to panel data. However, we use cross-sectional data, so this method cannot be applied. In addition, some of the variables we have defined as framing factors do not change over time. Their effects could, therefore, not be analysed within a fixed effects regression model. However, we understand that the different events and the characteristics associated with the events (such as the locations) could have an influence that we do not capture in our model, which could be why our model has low R-squared values. To address this issue, we introduce event-specific dummy variables when analysing the framing factors through regression analysis. This allows us to identify event-specific effects and to improve the r-square slightly. This resulted in changes within Table 7 on page 17 and the results chapter, see page 15, line 323 ff.

According to (b): We tested for heteroscedasticity and found that it was confirmed. Therefore, we applied bootstrapping.

According to (c) The regression analysis of the framing factors shows low R-squared values. This is a known problem in psychological research. It is due to the fact that people are very different, but they do not participate in interviews that last longer than 30 minutes, making it impossible to include all personal and contextual factors (Grothmann & Reusswig, 2006). |

| | potential way to improve the fitness of your model may be to account for insurance claim data (this data is difficult to obtain at the granular level due to privacy issues, though). It could be interesting to insert the amount of damages faced by these households in their attitudes and their reactions to the events. They may have experienced flood events first-hand, but if the damages were not so consistent they may have been left unaltered by the events. | According to (d) We don't get access to insurance claims, but we have compared mean losses from our survey with mean insured losses reported by the German Association of Insurers. |
|---|---|---|
| | To conclude, I hope you fill find these comments useful and I wish you good luck with the rest of your work! | Thank you very much again. |

**References**

Andrasko, I. (2021). Why People (Do Not) Adopt the Private Precautionary and Mitigation Measures: A Review of the Issue from the Perspective of Recent Flood Risk Research. *Water*, *13* (2). https://doi.org/https://doi.org/10.3390/w13020140

Bird, D., King, D., Haynes, K., Box, P., Okada, T., & Nairn, K. (2013). *Impact of the 2010–11 floods and the factors that inhibit and enable household adaptation strategies*. https://nccarf.edu.au/wp-content/uploads/2019/03/Bird_2013_Floods_household_adaptation_strategies.pdf

Brockie, L., & Miller, E. (2017). Understanding Older Adults' Resilience During the Brisbane Floods: Social Capital, Life Experience, and Optimism. *Disaster Medicine and Public Health Preparedness*, *11*(1), 72-79. https://doi.org/10.1017/dmp.2016.161

DeStatis. (2014). *Ergebnisse des Zensus am 9. Mai 2011* [table]. Statistische Ämter des Bundes und der Länder.

Erdlenbruch, K., & Bonté, B. (2018). Simulating the dynamics of individual adaptation to floods. *Environmental Science & Policy*, *84*, 134-148. https://doi.org/https://doi.org/10.1016/j.envsci.2018.03.005

Gilligan, C., Kypri, K., & Bourke, J. (2014). Social Networking Versus Facebook Advertising to Recruit Survey Respondents: A Quasi-Experimental Study. *JMIR Res Protoc*, *3*(3), e48. https://doi.org/10.2196/resprot.3317

Grothmann, T., & Reusswig, F. (2006). People at Risk of Flooding: Why Some Residents Take Precautionary Action While Others Do Not. *Natural Hazards*, *38*(1), 101-120. https://doi.org/10.1007/s11069-005-8604-6

Houston, D., Werritty, A., Ball, T., & Black, A. (2021). Environmental vulnerability and resilience: Social differentiation in short- and long-term flood impacts. *46*(1), 102-119. https://doi.org/https://doi.org/10.1111/tran.12408

Kapp, J. M., Peters, C., & Oliver, D. P. (2013). Research Recruitment Using Facebook Advertising: Big Potential, Big Challenges. *Journal of Cancer Education*, *28*(1), 134-137. https://doi.org/10.1007/s13187-012-0443-z

Lindell, M. K., & Perry, R. W. (2012). The Protective Action Decision Model: Theoretical Modifications and Additional Evidence. *Risk Analysis*, *32*(4), 616-632. https://doi.org/10.1111/j.1539-6924.2011.01647.x

Prentice-Dunn, S., & Rogers, R. W. (1986). Protection Motivation Theory and preventive health: Beyond the Health Belief Model. *Health Education Research*, *1*(3), 153-161. https://doi.org/10.1093/her/1.3.153

Shaver, L. G., Khawer, A., Yi, Y., Aubrey-Bassler, K., Etchegary, H., Roebothan, B., Asghari, S., & Wang, P. P. (2019). Using Facebook Advertising to Recruit Representative Samples: Feasibility Assessment of a Cross-Sectional Survey. *J Med Internet Res.*, *21*(8), e14021. https://doi.org/10.2196/14021

Thieken, A. H., Bubeck, P., Heidenreich, A., von Keyserlingk, J., Dillenardt, L., & Otto, A. (2023). Performance of the flood warning system in Germany in July 2021 – insights from affected residents. *Nat. Hazards Earth Syst. Sci.*, *23*(2), 973-990. https://doi.org/10.5194/nhess-23-973-2023

Wasserhaushaltsgesetz vom 31. Juli 2009 (BGBl. I S. 2585), (2009).

---

## Author Response (AR2)

Dear Associate Editor,

on behalf of all the authors, I would like to thank you for the constructive comments and criticism received on our manuscript entitled 'Individual Flood Risk Adaptation in Germany: Exploring the Role of Different Types of Flooding'. We believe that in the current revision we have addressed the comments raised and that in doing so our revised manuscript is now more suitable for publication in Natural Hazards and Earth System Sciences.

| No. | Comment | Answer |
|---|---|---|
| 1 | The abstract was too lengthy. Only the most important findings should be included. | We have shortened the abstract with a focus on the key findings. |
| 2 | The research gaps should be further elaborated. | Thank you for pointing this out. In the revised version we are now going into the research gap in more detail:

Page 2, line 53f: added text: . "There has not yet been a study focusing on adaptive behaviour and its drivers in the context of different types of flooding to identify and analyse factors influencing the motivation to adapt."

Page 4, line 100ff: added text: "While past research has analysed factors that influence adaptive behaviour solely in the context of one specific type of flooding ((Bubeck et al., 2020; Dillenardt et al., 2022; Grothmann & Reusswig, 2006) there is a lack of research about how those influencing factors are differing among flood types. " |
| 3 | Try to avoid the descriptions of "we did...", and change it to passive voice. | We have converted all formulations with "we" into an impersonal formulation. |
| 4 | Change the caption of Figure 2, as the cross-referencing should be avoided. A figure or table should be able to explain itself. | The caption of Figure 2 reads: "Figure 2: Simplified illustration of the survey designs A - D used to contact Flood-affected households in Germany" and does not contain any cross-referencing. The numbering A-D refers to the |

| | | |
|---|---|---|
| | | numbering shown in the figure itself. This illustration is therefore self-standing. |
| 5 | All the abbreviations should be explained at their first shown-up places, such as "SPAA", "ANOVA", .... Some abbreviations that occurred only once should be deleted. | We are unsure what you mean by the abbreviation SPAA as it does not appear in the paper. We suspect that you might mean SPSS instead. The fact that this is the name of the IBM software package used is already explained in line 169, page 7.We have now also added the written-out form of the name in line 170f, page 7.

 The abbreviation ANOVA was explained in line 173, page 7.

 All other abbreviations have been checked: They have already been explained when they were first used in the text. |
| 6 | There are many grammars, the writing should be significantly improved. | We have proofread the text. |